# Distributary development in a 21st century river: The evolution of Neptune Pass and its delta, the largest new offshoot of the Mississippi River

**Alexander S. Kolker** [1,2�]*, **H. Dallon Weathers**[3,4�], **Christy Swann**[5,4�], **Alisha A. Renfro**[6�]

**1** Louisiana Universities Marine Consortium, Chauvin, Louisiana, United States of America, **2** The Coastal Climates Institute, New Orleans, Louisiana, United States of America, **3** Delta Geo-Marine, New Orleans, Louisiana, United States of America, **4** Department of Earth and Environmental Sciences, University of New Orleans, New Orleans, Louisiana, United States of America, **5** RCOAST, New Orleans, Louisiana, United States of America, **6** Gulf Program National Wildlife Federation, New Orleans, Louisiana, United States of America

l These authors contributed equally to this work.

* akolker@lumcon.org

## Abstract

The development of distributaries in large river deltas plays an important role in the geology, hydrology, and ecology of the coastal ocean, as large rivers are a dominant mechanism by which particulate, suspended, and dissolved material is delivered from the continents to the global ocean. And yet, there is relatively little, near-real time observational data on the development of distributaries in large river deltas -- in part because the development of modern observation coincides with an era when rivers have been controlled by large engineering projects (i.e., the 20th and 21st centuries). This article reports on Neptune Pass, the largest new distributary to form in the Mississippi River in nearly a century. It developed between 2019 and 2021 when a small canal rapidly expanded by at least an order of magnitude. The system now carries about 15–17% of the flow of the Mississippi River, $> 3,000$ m$^3$ s$^{-1}$ when the Mississippi River is at moderately high flows. This is comparable to the 10th largest river in North America and the 100th largest river on Earth. Neptune Pass is building a delta, and this study sought to examine whether this delta is comprised largely of material eroded from the Neptune Pass (redistributed sediment hypothesis), or includes material recently derived from the Mississippi River (new sediment hypothesis). These hypotheses were tested using a combination of marine-geophysical surveys, remote sensing techniques, and sediment core collections. Results indicate that the delta in Quarantine Bay was 56–79% larger than the material excavated from Neptune Pass, corroborating the new sediment hypothesis, and indicating that it is a net land building system. These findings provide key insights that are critical to the restoration and safe management of the Mississippi River and its delta, the largest system of its kind in North America.

**Data availability statement:** All data are available from the Louisiana Coastal Protection And Restoration Authority, by making a data request here: https://cims.coastal.louisiana. gov/DataRequest.aspx Data Are Also Available At Pangaea.edu

**Funding:** This project was funded, in part, through a sub-contract with the Louisiana Coastal Protection and Restoration Authority, who was funded under Award No. GNTCP18LA0035 from the Gulf Coast Ecosystem Restoration Council (RESTORE Council), and through multiple contracts with the National Wildlife Federation. This work represents contract #20220831Task Order No.4 from the Louisiana Coastal Protection And Restoration Authority to the Louisiana Universities Marine Consortium. The data, statements, findings, conclusions, and recommendations are those of the authors and do not necessarily reflect any determinations, views, or policies of the RESTORE Council. We acknowledge that Dr. Alisha Renfro from the National Wildlife Federation contributed to this article in her capacity as a scientist and scholar. Links to project sponsors can be found here: https://coastal.la.gov; https://www.restoretheg-ulf.gov; https://www.nwf.org

**Competing interests:** The authors have declared that no competing interests exist.

# 1 Introduction

## 1.1 Introduction to Neptune Pass

The development of distributaries in large river deltas plays an important role in the geology, hydrology, and ecology of the coastal ocean [1–4]. Large rivers are a dominant mechanism by which particulate, suspended, and dissolved material are delivered from the continents to the global ocean. At the mouth of many of the world's largest rivers is a delta- a progradational sedimentary deposit. Deltas provide habitat for ecologically and economically important species, recycle nutrients, and often serve as a locus for human activities [5,6]. Changes to river mouths, either through avulsions (major changes) or the formation of smaller distributaries (partial changes) can reroute the flow of freshwater in the coastal ocean, change the geology of coastal land/bedforms, and alter the biogeochemistry and ecology of coastal waters [7–9].

Changes to river outlets have the potential to impact the people whose lives or livelihoods depend on coastal waters [10]. In large river deltas, such shifts can impact sediment supply: changing patterns of land gain/loss, which is a critical parameter influencing coastal resilience during times of climate change and sea level rise. For example, changes to the directionality of river mouths can alter the distribution of freshwater and nutrients which could affect coastal fisheries and the people who depend on them.

Despite widespread recognition in the scientific and sustainability communities that changes to river mouths can be geologically and societally important, there are relatively few modern observational studies of them - particularly in North America. This gap exists because the development of modern coastal observation methods (i.e., water-level gauges and remotely sensed imaging) is largely contemporaneous with an era (roughly the 20th century) of increased coastal engineering that limited the capacity for river mouths to shift. However, there have been more extensive studies based on analyses of historical maps, sediment cores, numerical, and physical modeling [11–14]. Thus, studies like this present one, which provide near-real time observational evidence of a large distributary and an associated delta forming can be of significant interest to multiple disciplines including hydrology, geology, ecology, biogeochemistry, and coastal sustainability.

Exploring recent developments near the mouth of the Mississippi River provides a unique and societally important opportunity to understand river mouth development in the coastal ocean. In the lower Mississippi River Delta a new crevasse developed between 2019–2021. This channel, now known as Neptune Pass, is located along the eastern bank of the Mississippi River 39 km above Head of Passes (nominally the river's mouth; Fig 1). Its discharge likely exceeds 4,000 m³ s⁻¹, at high flow [15], which amounts to about 15% of the total flow of the Mississippi River- the largest river in North America. It is building a delta in Quarantine Bay in lower Brenton Sound (one of the large bays of the Mississippi River Delta) that is one of the largest new deltas in North America. This article describes the formation of Neptune Pass and its associated delta and tests a hypothesis about the nature of the delta deposit which has management implications for the Mississippi River and its delta.

As a new and large distributary system, Neptune Pass has the potential to both build large areas of land and impact navigation in the Mississippi River -- both of which are critically important societal needs. The Mississippi River Delta has lost about 25% of its land area over the last century, which accelerated flood risks for coastal residents and reduced habitat for numerous ecologically and economically important species [1,16,17]. Moreover, the Mississippi River is the largest pathway by tonnage for commercial shipping in the United States [17]. Alterations to the river channel could create hazards for vessels, (about half of which carry fossil energy or petrochemicals,) potentially creating supply chain problems that could impact local, national, or global commerce [4].

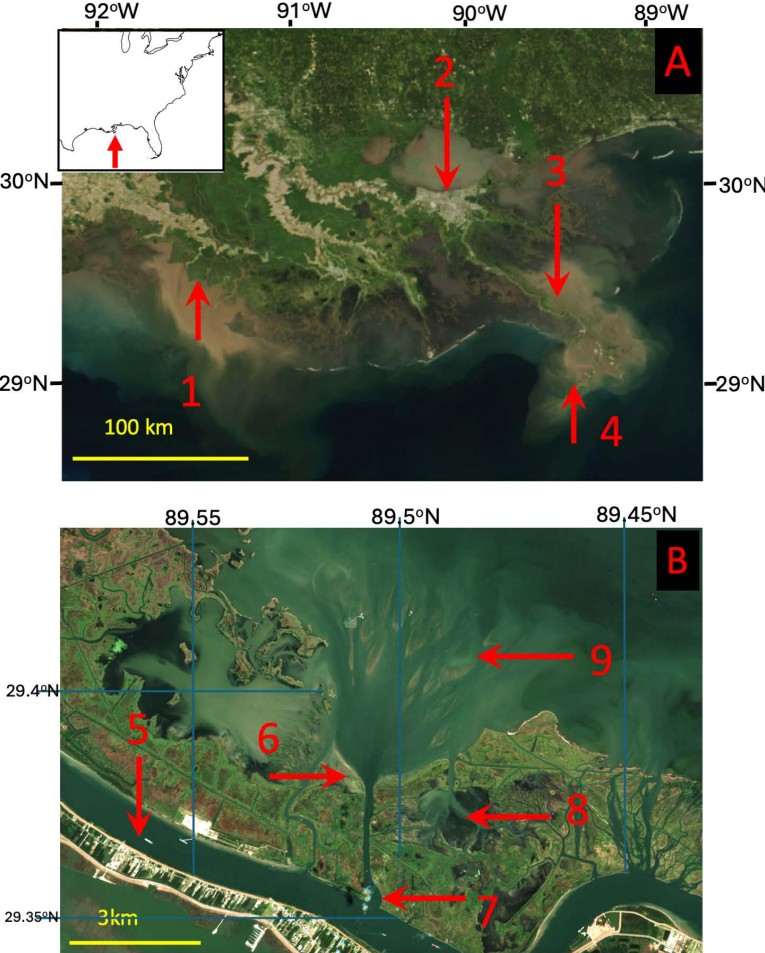

**Fig 1. Map of coastal Louisiana with major features noted.** A. Large scale view with the following locations noted: 1. The Wax Lake Delta of the Atchafalaya River. 2 New Orleans. 3. Neptune Pass. 4. Southwest Pass of the Mississippi River. Image Source: NASA-Modis. B. Close in view with the following locations noted; 5. The Mississippi River. 6. Ducks Unlimited Terraces. 7. Entrance to Neptune Pass. 8. Bay Denesse. 9. Quarantine Bay and the Quarantine Bay Delta. Satellite Image Source A. NASA Worldview (worldview.earthdata.nasa.gov), part of the NASA Earth Science Data and Information System (ESDIS;). Image date: April 22, 2022.Inset US Census: B. Copernicus Sentinel-2 Data. Image date: October 24, 2022.

The impetus for the present study is the rapid expansion of Neptune Pass which started in about 2020. In 2022, satellite images revealed Neptune Pass was leading to delta growth [15]. Preliminary evidence suggested that Neptune Pass may be leading to one of the largest deltaic land building events in Louisiana (and potentially beyond) since the emergence of the Wax Lake and Atchafalaya River Deltas in the 1970s [18–20]. Additionally, other reports indicate that new river currents formed near Neptune Pass and that shoals were developing downstream of it, potentially leading to navigation obstacles and hazardous shipping conditions in the Mississippi River [21,22]

The emergence of deltaic lands leads to important questions about the source material for this delta, and its long-term sustainability. Here we propose two new hypotheses regarding the source of the material that is contributing to land formation. One hypothesis holds this land represents "new" deltaic land building with sediments sourced from the Mississippi River. Under this scenario, delta evolution would follow a progression similar to natural crevasse

splay evolution and to the river diversions that are part of Louisiana's Coastal Master Plan and ongoing restoration activities [17]. If this hypothesis were shown to be valid, sustainable land development in the years and decades ahead is likely because the system receives a relatively steady supply of sediment. An alternate hypothesis posits the material comprising the delta in Quarantine Bay is "redistributed" material - sourced from the scour that occurred during the expansion of Neptune Pass. This hypothesis suggests future land building would be minimal and dependent largely on further erosion of Neptune Pass's main channel. It should be noted that these hypotheses are not mutually exclusive and represent end-members on a spectrum of sediment input possibilities. These hypotheses can inform coastal zone management, and the broader geoscience community, who are both interested in sediment transport and deposition in this large river delta.

The major purpose of this study is to examine these competing hypotheses while also quantifying the system in enough detail to provide useful insights into the processes driving its evolution. A pragmatic way to examine these hypotheses is through a *volume-balance* or *mass-balance* approach. The quantity (volume and/or mass) of sediment removed from Neptune Pass can be compared to the quantity of sediment deposited in the Quarantine Bay receiving basin. If the quantity of newly deposited sediments in Quarantine Bay exceeds the quantity of material extracted, this would support the "new land" hypothesis. If the volume is less, then the "redistributed" hypothesis would hold more weight.

This study was conducted using marine geophysical tools, sediment core collections, aerial drone-based surveys, and satellite image analysis. While the primary goal was to quantify the amount of material scoured from Neptune Pass and deposited in Quarantine Bay, the approach was broad enough to generate data that provide a geological context for the evolution of Neptune Pass and associated landforms. Ideally, the information will be useful to a broad range of geoscientists, as well as people at government agencies who are making environmental-management decisions about North America's largest river and delta. Residents, community groups, and non-governmental organizations who want to contribute to public discussions about these decisions are likely to find this information useful too.

## 1.2 Study location

Neptune Pass is located along the southeast side of the lower Mississippi River, approximately 39 km above Head of Passes (Fig 1). It is near, but on the opposite side of the river from, the town of Empire, Louisiana. Neptune Pass is in a reach of the Mississippi River that extends from the Bohemia Spillway to Head of Passes where levees on the river's eastern bank are not maintained -- though rock walls and jetties help maintain overall channel stability in this region. In this reach, channels connect the Mississippi River to Breton Sound, many of which formed between 1972 and the present day. Neptune Pass is the newest and largest of these.

Other notable channels in the area connecting the Mississippi River to parts of Breton Sound include the following:

- Mardi Gras Pass - a channel that formed in winter 2012 with a discharge of about 300 ~ 600 $m^3$ $s^{-1}$ [45];

- The Ostrica Locks complex - a defunct and now closed set of navigation locks that is adjacent to an open and unregulated channel with a discharge 300 ~ 600 $m^3$ $s^{-1}$ [45];

- The Fort St Philip Crevasse complex - a series of channels that formed after a large flood in 1973. Individual channels in the Fort St Philip Crevasse Complex have a discharge 100–600 $m^3$ $s^{-1}$, and the entire system has a discharge of about 3,000 $m^3 s^{-1}$ [44].

In addition to these relatively large channels, other smaller ones exist in this region. The area between the Ostrica Locks and the end of the named sections of channel is referred to by the US Army Corps of Engineers as the "Olga" reach of the river and will be referred to as such in this article.

Neptune Pass discharges into Quarantine Bay, a part of the Breton Sound Hydrologic Basin, on the eastern flank of the Mississippi River Delta. Breton Sound has lost about 42% of its wetlands over the past century [16]. Multiple factors contribute to this land loss which include: the construction of canals, reductions in the delivery of sediment and freshwater to wetlands from levee construction, high rates of subsidence, storms, relative sea level rise, and oil spills [17,24]. Breton Sound is the proposed site of the Mid-Breton Sediment Diversion that was part of Louisiana's Coastal Master Plan in 2017 [17]. This diversion project seeks to restart natural land building processes by diverting up to 1,416 $m^3$ $s^{-1}$ of freshwater and its associated sediment load from the Mississippi River to Breton Sound [17]. As described in more detail in the discussion section of this article, the Neptune Pass/Quarantine Bay system is similar to the Mid-Breton Diversion, though Neptune Pass's maximum discharge is larger ($\sim$ 3400 vs 1416 $m^3$ $s^{-1}$ maximum), and its flow is not gated. Neptune Pass is also about 70 river km downstream, and on the other side of the river from the Mid-Barataria Sediment Diversion, a nearly $3 billion coastal restoration project comparable to, but slightly larger than, the Mid-Breton Sediment Diversion.

Prior to 2019, the system now known as Neptune Pass was a small channel between the Mississippi River and Breton Sound (Fig 2). The system was unnamed but will be referred to in this article as "Proto-Neptune Pass." One survey of Proto-Neptune Pass, conducted during high flow of the Mississippi River in 2016, measured a discharge of approximately 300 $m^3$ $s^{-1}$ [15,23]. Satellite images show the system rapidly expanded between 2019 and 2021 (Fig 2). The pass increased from approximately 30–50 m wide and about 3–6 m deep in 2016 to an average of 200 m wide (and wider near the mouth) with depths reaching up to 30 m at present [15,23]. As the size of the channel increased, the discharge increased by roughly an order of magnitude and possibly more (Table 1).

### February 7, 2019

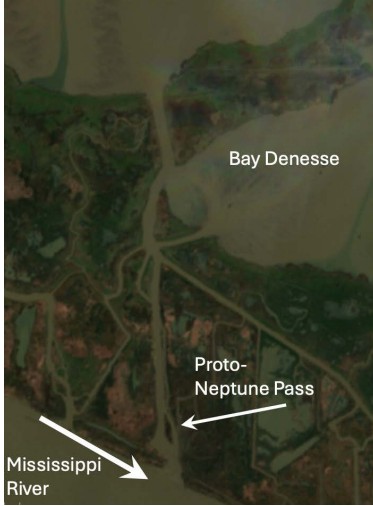

### March 8, 2021

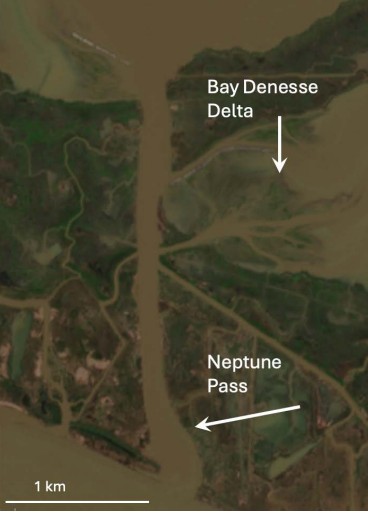

**Fig 2. Remotely senses images show the rapid expansion of Neptune Pass.** Source Copernicus Sentinel-2 Data for February 7, 2019 and March 8, 2021.

**Table 1. Results from April 2022 and May 2024 Discharge Surveys.**

| Survey Location or Calculation Information | April 24, 2022 | February 16, 2024 |
|---|---|---|
| Mississippi River at Belle Chasse Gauge (m³s⁻¹) | 22,080 | 22,055 |
| Mississippi River Above Neptune Pass (m³s⁻¹) | 20,290 | 20,531 |
| Neptune Pass (m³s⁻¹) | 3,360 | 3,205 |
| Mississippi River Below Neptune Pass, Fort St Philip and Other Nearby Crevasses (Olga Downstream) (m³ s⁻¹) | 12,320 | 14,142 |
| Total Eastward Flow (m³ s⁻¹) | 9,761 | 7,913 |
| % of River Discharge at Neptune Pass (Relative to Belle Chasse) | 15% | 15% |
| % of River Discharge Eastward (Relative to Belle Chasse) | 36% | 29% |

Results from April 2022 and May 2024 discharge survey conducted by this team. The Belle Chasse data from US Geological Survey's Mississippi River gauge #07374525.

Neptune Pass's flow is currently unregulated, though there have been attempts to stabilize the channel. In 2022, the U.S. Army Corps of Engineers (USACE) placed rocks along the channel's bottom and at the entrance to Neptune Pass where it connects to the Mississippi River. In 2023, the USACE placed rocks along the southern edge of the channel's entrance. The USACE has plans to further reduce the flow of the system, by placing a rock wall near the entrance of Neptune Pass, and installing sediment retention devices near its exit into Quarantine Bay [22]. The data and analyses presented herein can be used to inform comments on the USACE proposal and inform the public at large.

## 2 Methods

### 2.1 Discharge

To understand the discharge of Neptune Pass and its potential impacts on the overall flow distribution in the lower Mississippi River, a series of hydrographic surveys were conducted by this team on May 24, 2022 and on February 16, 2024. These surveys were conducted using an Acoustic Doppler Current Profiler (ADCP) mounted to the side of a small vessel (the 6m M/V New Delta). Surveys were conducted upstream of Neptune Pass, within Neptune Pass, and downstream of the entire Fort St Philip/Neptune Pass reach of the Mississippi River (Fig 11). Additionally, smaller surveys were conducted in several of the crevasses of the Fort St Philip Crevasse complex.

This project also compiled long-term data on discharge in the Mississippi River from multiple outside sources. These sources include: 1) vessel-based survey of discharge in the lower Mississippi River and its outlets collected by the US Army Corps of Engineers using methods similar to those used by this group. 2) Measurements of stage and discharge at the US Geological Survey's gauge at Belle Chasse, Louisiana. Finally, data were compiled on discharge and stage at the USGS Belle Chasse gauge (available at waterdata.usgs.gov, gauge number 07374525) and data on water levels at other nearby stations in the Mississippi River collected by US Army Corps of Engineers (available at rivergauges.com).

### 2.2 Neptune Pass channel bathymetry survey via multi-beam sonar

To understand the development of Neptune Pass, a multi-beam bathymetric survey was conducted in Neptune Pass on June 2, 2022 using the research vessel the R/V Penland. The survey was conducted using Reson SeaBat 7125 SV2 High Resolution Multibeam Sonar System equipped with a Position and Orientation System for Marine Vessels system (POS MV). Qinsy

software was used to collect the data; standard hydrographic calibrations were performed via a sound velocity cast, and a patch test was used to calibrate the sensors.

## 2.3 Quarantine Bay bathymetry survey via single-beam sonar

A single-beam bathymetric survey was conducted in Quarantine Bay in September and November of 2022 to determine seabed elevations and the morphology of subaqueous deposits. Survey lines were informed by Copernicus Sentinel-2 satellite images of Quarantine Bay captured in September and October 2022 ([Fig 4]). They included areas where landforms were likely to be emerging and extended beyond these areas to ensure coverage of the distal extent of the depositional system.

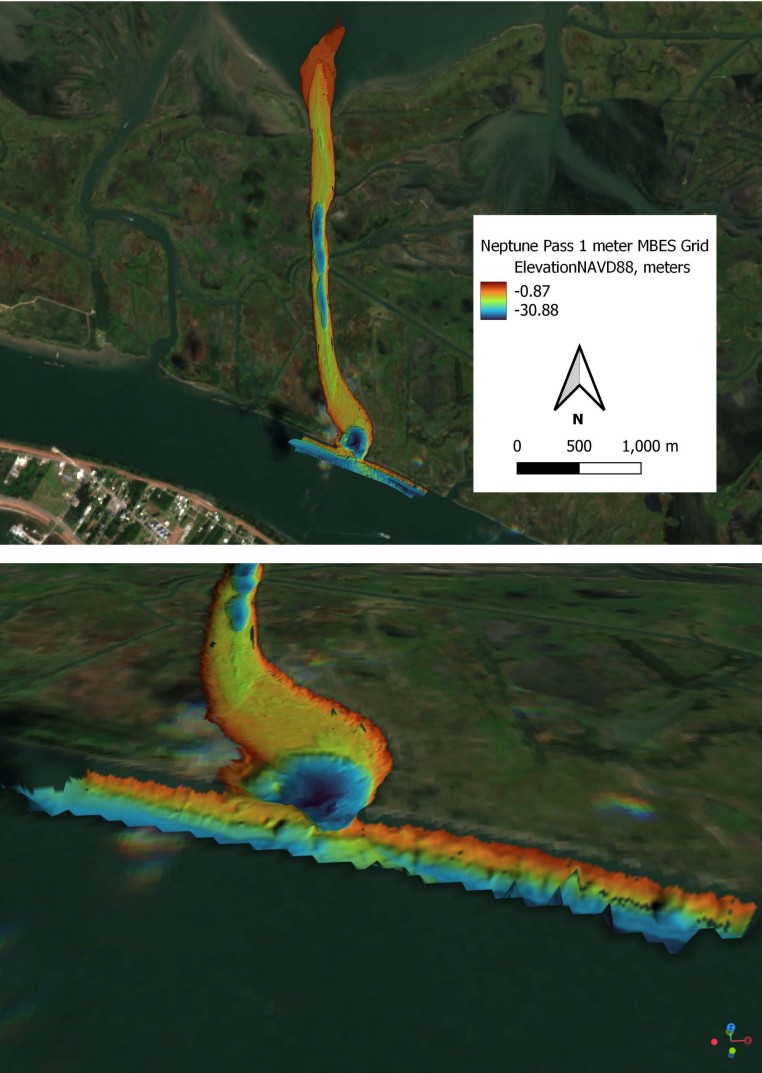

**Fig 3. Multibeam bathymetry of Neptune Pass, which was used to determine the volume of scour in the Neptune Pass channel for the sediment budget.** A. Plan view. B. Oblique view, as would be seen from an observer located above the Mississippi River looking towards the entrance to Neptune Pass. Note the uneven ledge and large scour hole at the entrance to Neptune Pass and the steep channel walls in the main reach of the pass -- all are indicative of an active erosional environment. Image Source: Copernicus Sentinel-2. Image Date: October 24, 2024.

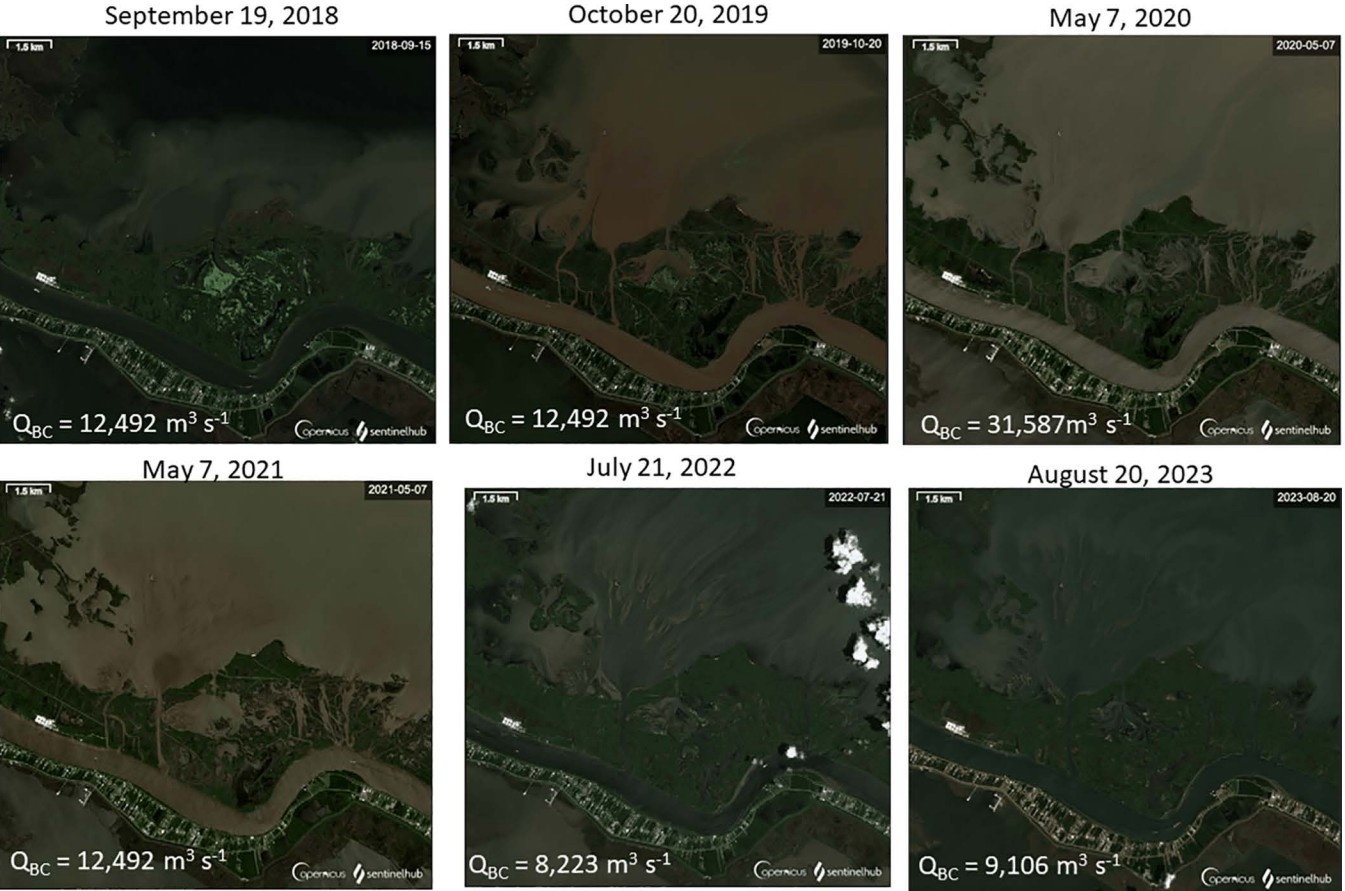

**Fig 4. The evolution of the Neptune Pass plume and the Quarantine Bay Delta between 2018 (pre-expansion) to near the present day (August 20, 2023).** The discharge at Belle Chasse, Louisiana (USGS Gauge 070374525) is noted in the lower left corner of each image. The images were selected to show both the temporal evolution and a diversity of river stages. The May 7, 2020 image relates to a discharge of 31,587 $m^3$ $s^{-1}$, which is near the maximum allowable flow in the lower Mississippi River (35,400 $m^3$ $s^{-1}$), and the July 21, 2022 and August 20, 2023 near historic low flow values. Image Source: Copernicus Sentinel-2 Data; Discharge data source: waterdata.usgs.gov.

Surveys were conducted on the University of New Orleans' research vessel, the R/V Mud-lump (Fig 5), and employed an Odom Hydrotrac single-beam system, mounted on a pole operating at 200 kHz. A Trimble R8 real-time kinematic (RTK) GPS unit was mounted on the pole 1.86 m directly above the transducer. To correct for the impacts of boat movements on the survey, a Teledyne Dynamic Motion Sensor (DMS-25) was mounted to the survey pole to measure heave, pitch, and roll. Survey data were integrated, collected, and processed in the Hypack software environment (Fig 5).

## 2.4 Quarantine Bay sub-bottom survey via CHIRP seismic

A sub-bottom seismic survey was run concurrent to, and along the same survey lines as, the single-beam bathymetric survey. An Edgetech 3100-P CHIRP sub-bottom profiling system was used along with an SB-216 towfish to collect sub-bottom data. This system emits acoustic energy in a sweeping pulse in the frequency range of 2–16 kHz and collects returns in a transducer array in the towfish to generate sub-bottom imagery. Seafloor penetration is typically between 1–5 meters, depending on the water depth and the thickness, particle size, and bulk

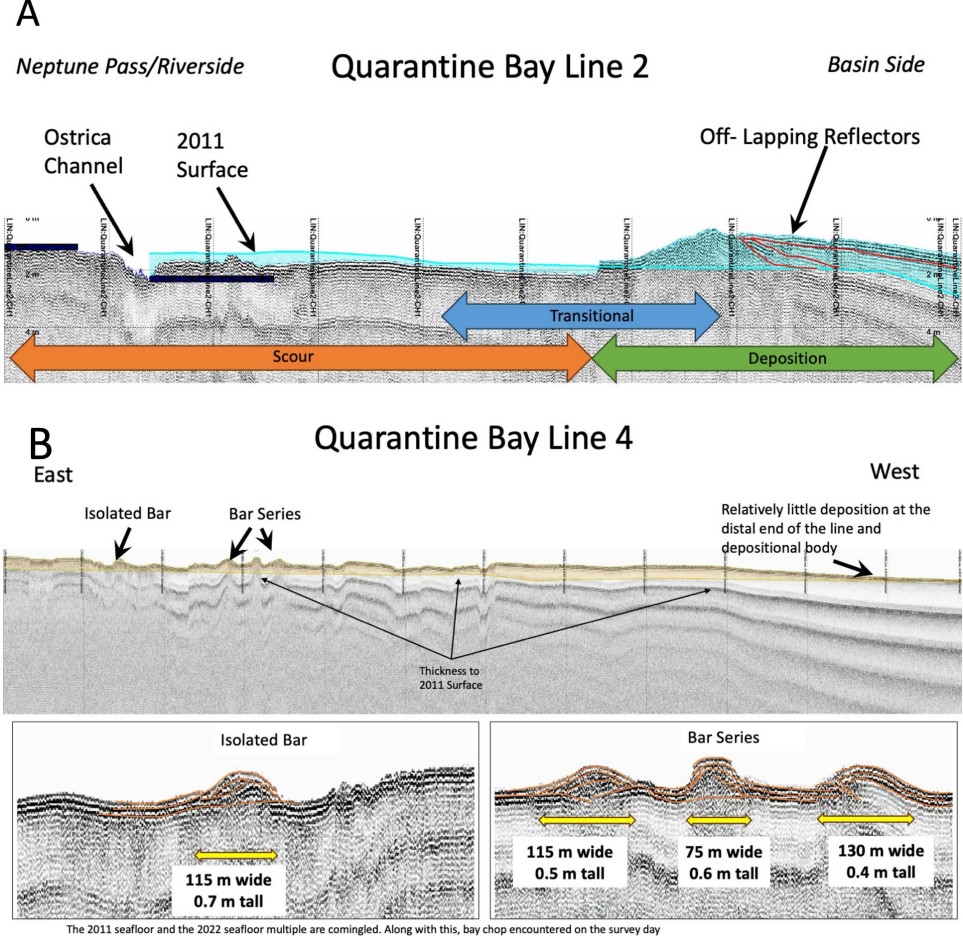

**Fig 5. CHIRP sonar data for the Quarantine Bay Delta. A.** A line that extends along-dip from the mouth of Neptune Pass (the riverside) to the bayside end of the Quarantine Bay Delta. This annotated image shows areas of erosion -- near the mouth, along with a bar, and multiple layers of sediment deposition. This image was also used to help determine the zones of erosion, transition, and deposition used in the sediment budget. **B.** An along-strike image of the Quarantine Bay Delta, with mouth bars and their spatial scale clearly noted.

density of sediment (Fig 6). The sub-bottom investigation was conducted to image the sedimentary structures and geometry of the deposit, including the emerging landforms mentioned in Section 3.1. The track lines of the CHIRP sonar survey were the same as the single-beam bathymetric survey presented in Fig 6.

## 2.5  Wetland topography and vegetation survey via Aerial Drone

Airborne drone surveys were launched on October 13, 2022 and March 28, 2023 using a DJI Matrice 300 RTK system mounted with a Zenmuse L1 gimbal payload which houses both a LiDAR (Light Detection and Ranging) and a high-resolution true color, red, green, and blue (RGB) camera. The LiDAR unit sends out pulsed light and records the time for pulses of light to return to the sensor, as well as the amount of light returning to the sensor for each pulse. These two methods produce a high-resolution 3D point cloud of the scanned area. Coupling the LiDAR with the RGB pixels from the sensing camera, true-color 3D topographic maps are

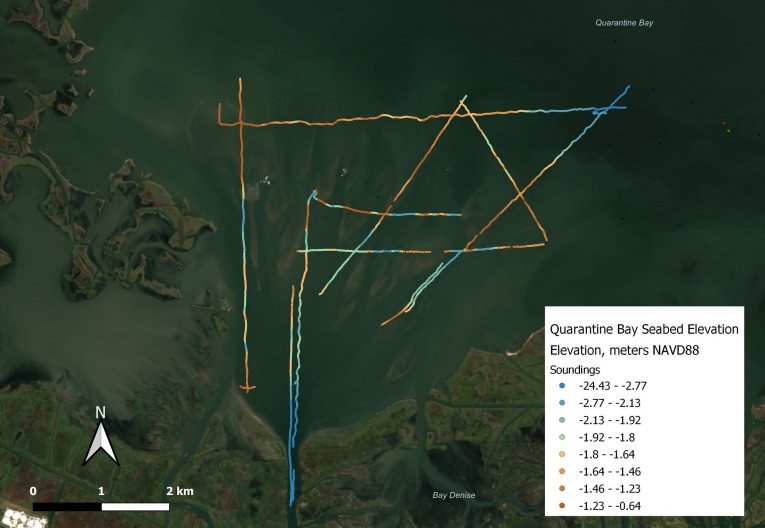

**Fig 6. Track lines and bathymetry data for Quarantine Bay single-beam survey.** Note: these track lines were also used for the CHIRP sub-bottom profiler survey. Background image source: Copernicus Sentinel-2 Data Service for October 24, 2022.

produced with classified surface characteristics such as vegetation type and density from the amount of light that is absorbed and returned from each LiDAR pulse.

The programs DJI Terra (https://enterprise.dji.com/dji-terra) and CloudCompare (https://github.com/CloudCompare/CloudCompare/releases/) were used to build digital elevation models (DEM) from the dense 3D point clouds to create continuous surfaces of the scanned area. Data were georeferenced using GPS RTK data points in QGIS for each of the scanned areas to resolve real-world coordinates for each LiDAR pulse and camera pixel. Data were processed to produce both mosaics of Visible Light and True Color imagery as well as DEMs, using the 3D topography from LiDAR.

The drone surveys covered three primary areas (Fig 1):

- Ducks Unlimited Terraces. A region of mudflats, vegetation, and marsh terraces located to the west of the region where Neptune Pass discharges into Quarantine Bay. The terraces, essentially linear features of mud and sand, were placed in this location by Ducks Unlimited in 2020 as part of their restoration program. These terraces will be referred to as the "DU Terraces" throughout this article.

- Bay Denesse and its newly formed delta. Bay Denesse is a ~ 3.5 km² bay to the east of the main channel of Neptune Bay. It is hydrodynamically connected to Neptune Pass by two primary channels and may receive additional sediment and water from ancillary flow paths. The primary flow pathways are connected to an emerging delta with a mixture of both herbaceous and woody vegetation, such as Black Willow (*Salix Nigra*) and Delta Duck Potato (*Sagittaria platyphylla*). Similar plant communities are found in other young deltas [20,25].

- The Quarantine Bay Delta. As described in the introduction, a large (~20–30 km²) delta is developing in Quarantine Bay. It has a series of shallow submerged and emergent landforms. These areas were evaluated for drone mapping.

## 2.6 Quarantine Bay Delta bathymetry: Pioneering satellite-derived bathymetry for the Mississippi River delta

To further understand the bathymetry of this shallow-submerged deltaic system, we developed a new method for satellite-derived bathymetry (SDB) in river deltas. SDB is an emerging technique to determine bathymetry using a remotely-sensed proxy correlated with observed depth [26,27]. SDB methods have been applied in coastal systems across the world -- though more often in clear water settings like coral reefs, and tropical embayments, where ocean color is related to water depth [26,27]. A review of the literature indicated that SDB has not been applied to settings in the Mississippi River Delta. As such, this study presents a novel method for determining landscape evolution in North America's largest delta -- and potentially other deltaic and muddy and sandy coastal environments.

This research team developed an algorithm to correlate Copernicus Sentinel-2 visible imagery to in-situ measurements of seabed elevation collected by the single-beam surveys. More specifically, the algorithm relates the pixel intensity ($P_R$) of the red band (band 4) of a Copernicus Sentinel-2 image collected on October 24, 2022 to water depths collected during the October 2022 surveys (Section 3.1, Fig 7, Fig 9). The algorithm functions on the concept that light is reflected from mouth bars that are in shallow water relatively strongly, whereas light is reflected from the seafloor in deep water relatively weakly. The red band was used because empirically it had the highest correlation to bar morphology. It was further hypothesized that the red band reflects light from sands and oxidized sediments, which can have an orange or reddish color. The blue and green bands used in SDB methods in other settings (clear water), were less effective in Quarantine Bay. The blue band works best in clear water -- which is not present in Quarantine Bay, while the green band likely picks up algae -- which is obscured by suspended sediment in Quarantine Bay [26,27]. The October 24 image was chosen because this day was cloud-free, the mouth bars were clearly visible, and it was approximate to the dates of the bathymetric survey.

The predictions were limited to water depths from -0.5 m NAVD 88 to extinction depth of -3.0 m NAVD 88, which eliminated emergent land on the shallow end and depth associated with the deeper parts of the Neptune Pass channel on the deep end, which reduced potentially spurious variability in the dataset. The relationship was $Q_{depth} = 2.98(-1.13* P_R)$. It yielded an $R^2$ of 0.37 and a root-mean squared error of 0.32, indicating the relationship is strong, but not perfect (Fig 8). We suspect that a majority of the unexplained variability stems from the difference in the spatial resolution of the Copernicus Sentinel 2 imagery and the in-situ bathymetry; 10 m vs ~ 10 cm. Since the variability is on the order of $10^{-2}$ to $10^{-1}$ m in the vertical, and many sedimentary features in Quarantine Bay are on the order of $10^0$ meter thick, $10^1$ to $10^2$ wide, and $10^2$ to $10^3$ meters long, the overall scale of the variability is small relative to the information needed to calculate a sediment volume. Future research should seek to refine these methods, using increased in-situ and remotely sensed methods.

## 2.7 Depositional environments determined via sediment core collection

Sediment cores were collected in Quarantine Bay on June 12 and 13, 2023. The location of sediment cores was informed and selected based on water depth, CHIRP sub-bottom profile data, and satellite images. The goal of the core collection was to understand the shallow geology of Quarantine Bay in enough detail to evaluate the thickness of recently sediment deposited in that bay. Cores were collected with a Vibracorer, a commonly-used method for collecting sediment cores in the Mississippi River Delta and other coastal systems [28,29]. A total of 12 cores were collected ranging in length from about 2–4 m (Table 2). The core length was governed by the depth of refusal - the depth at which the corer cannot readily penetrate

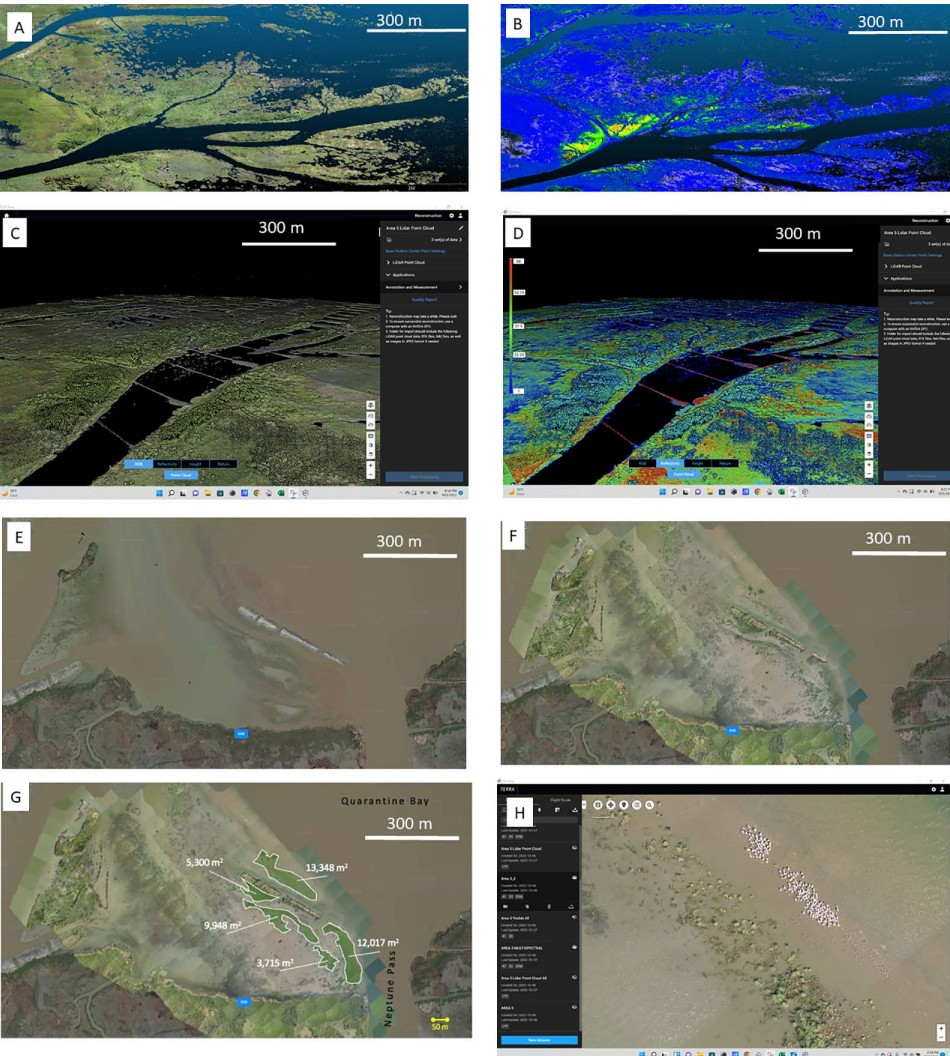

**Fig 7. Drone imagery.** Images of land development near Neptune Pass. A. Aerial drone imagery with true color and elevations (RGB+DEM) of the Bay Denesse Delta. B. Reflectivity values for this same delta. The reflectivity scale is relative, and bluer areas have lower reflectivity while the redder areas have less reflectivity. Note that the high grounds in the RGB+DEM images roughly correspond with areas of greater reflectivity, indicating higher, drier land that reflects a greater amount of light to the drone's sensor. C. RGB+DEM image of the entrance to the Bay Denesse Delta. D. Reflectivity image of the entrance to the Bay Denesse Delta. E. Overhead view of the DU Terraces from Google Earth on 1/17/2021. F and G. Drone images collected in October 2022 of the same terraces with the Google Earth image in the background. G shows areas of new growth highlighted, with the areas noted. H. Close up of this region showing white dots that are wetland birds, likely gulls, terns, or white pelicans.

further. This depth also roughly corresponds to the thickness of recently formed mouth bars in other nearby settings in the Mississippi River Delta [18,30]. The pipe diameter was 7.6 cm, and the pipe material was aluminum. While collecting sediment cores, the amount of compaction was determined by comparing the depth of the sediment in the core barrel to the depth of the water. Cores were processed and described at the University of New Orleans's Department of Geology and Geophysics [29,31]. The location of sediment cores is displayed in Table 2 and Fig 9, while a map with the location of single-beam sonar lines and the location of a relict oyster reef is displayed in Fig 10.

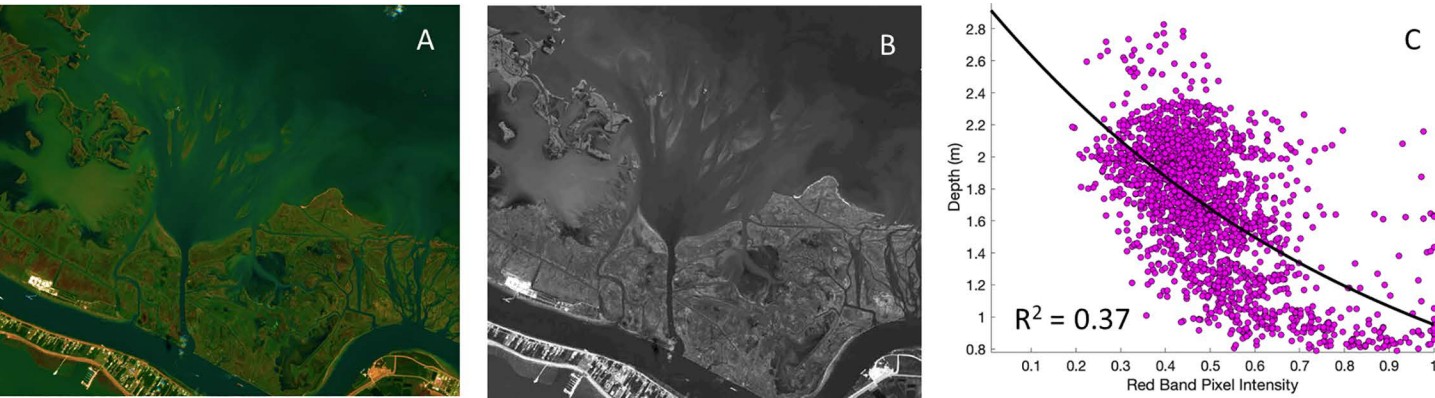

**Fig 8.  Information used to develop the satellite-derived bathymetry.** A. Copernicus Sentinel-2 true color image of the Quarantine Bay Delta on October 24, 2022. B. Copernicus Sentinel-2 image from October 23, 2022 in the red band (band 4). C. Relationship between in-situ measured water depth (Fig 7) and pixel intensity in the red band (b).

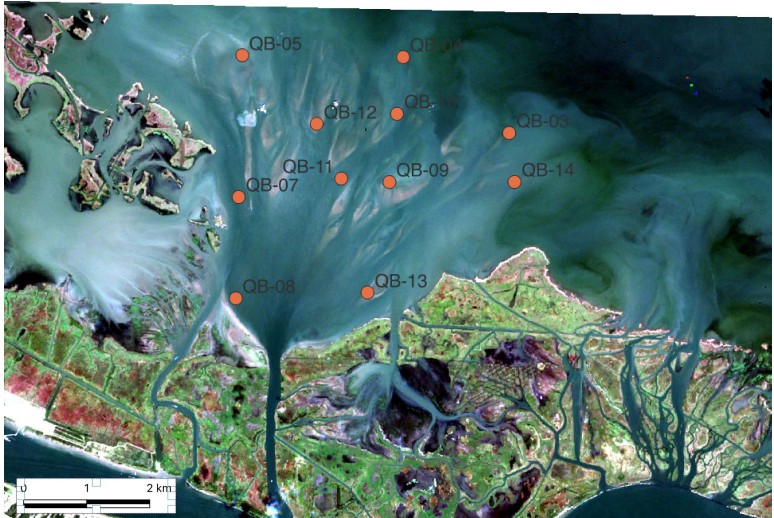

**Fig 9.  Location of sediment cores in collect in Quarantine Bay.** Image background: Copernicus Sentinel-2 Data Service. Image Date: October 24, 2022.

## 2.8  Calculation of sediment budget

One key component of this project was to calculate the amount of sediment deposited in Quarantine Bay, and to compare that to the amount of sediment scoured from Neptune Pass. This sediment budget can determine if the delta developing in Quarantine Bay was composed entirely of sediments scoured from Neptune Pass, or if some of the material in the Quarantine Bay Delta was sourced from recent sediment transport along the Mississippi River.

To calculate this sediment budget, the volume of sediment extracted from Quarantine Bay was determined via a difference approach. It compared the channel volume determined from a (pre-expansion) January 2016 survey to the channel volume determined from a (post-expansion) May 2022 survey [15,23]. The January 2016 survey used an Acoustic Doppler Current Profiler (ADCP) to measure depths at the head and mouth of Proto-Neptune Pass,

**Table 2. Core Locations and Key Information.**

| Core ID | Collection Date | Latitude | Longitude | Top Elevation NAVD88 (cm) | Core Length (cm) | Water Depth (cm) | Compaction (cm) | Notes |
|---------|-----------------|----------|-----------|---------------------------|------------------|------------------|-----------------|-------|
| QB-03 | 6/12/2023 | 29.416537 | -89.473728 | -92 | 326 | 162.6 | 29.2 | Lost 2 cm from bottom of core |
| QB-04 | 6/12/2023 | 29.427043 | -89.49138 | 19 | 329 | 188 | 200.7 | |
| QB-05 | 6/13/2023 | 29.426818 | -89.517666 | -97 | 340 | 101.6 | 45.7 | Core lost on 1st attempt. |
| QB-06 | 6/13/2023 | 29.432375 | -89.817549 | -119 | 301 | 147.3 | 100.3 | |
| QB-07 | 6/13/2023 | 29.406568 | -89.517763 | -146 | 284 | 172.7 | 53.3 | |
| QB-08 | 6/13/2023 | 29.392086 | -89.517867 | -113 | 161 | 119.4 | 12.7 | Short core because pipe sheared at refusal. |
| QB-09 | 6/12/2023 | 29.409169 | -89.49318 | -41 | 405 | 61 | 132.1 | |
| QB-10 | 6/12/2023 | 29.418928 | -89.492183 | -87 | 359 | 101.6 | 147.3 | |
| QB-11 | 6/13/2023 | 29.409523 | -89.501079 | -174 | 189 | 190.5 | -2.5 | Short core because pipe sheared at refusal. |
| QB-12 | 6/13/2023 | 29.417251 | -89.50527 | -29 | 357 | 45.7 | 15.2 | |
| QB-13 | 6/13/2023 | 29.39333 | -89.496384 | -27 | 343 | 35.6 | 48.3 | |
| QB-14 | 6/12/2023 | 29.409565 | -89.472691 | -87 | 318 | 106.7 | 134.6 | |

and channel dimensions were calculated based on linear interpolations. The 2022 survey used a multibeam sonar survey to measure the depth of the entire channel (Fig 3). Google Earth images suggest relatively little change in the channel between 2016 and 2019, directly before the opening of Neptune Pass (Fig 2).

The volume of material in the Quarantine Bay deposit was determined by creating an isopach map. In this map, the deepest part of the deposit was determined from survey lines collected by Louisiana's Department of Wildlife and Fisheries (LDWF) in 2011 [32]. This was the most recent and most accurate survey of Quarantine Bay conducted before the expansion of Neptune Pass. It was referenced to NAVD 88, and its validity was corroborated by sediment cores that have shells at the same locations and interpolated elevations, where the LDWF found oyster reefs (Fig 10). The LDWF survey was further adjusted for subsidence, assuming a rate of 22 mm yr$^{-1}$, as per literature values [33]. The higher end of the Quarantine Bay deposit was determined from the surveys presented herein.

For the sediment budget calculation, the bay was split into three main regions: an erosional zone (near the channel mouth), a transitional zone of mouth bars and channels, and a net depositional region of thicker deposits (Fig 10). These zones were determined from the CHIRP sonar -- which revealed patterns of erosion and deposition, as well as the in-situ and satellite-derived bathymetry, and sediment cores (Fig 6, Fig 10). These data reveal overall morphological patterns in the bay, while the sediment core data further corroborate patterns and volumes of deposition and erosion. In each of these three zones, the mean depth change (accounting for subsidence and adjusted to NAVD 88) between the 2011 and 2022 surveys was determined, and the volume was calculated from the area of each zone.

To further evaluate the amount of sediment scoured and deposited, the total mass of sediment of scoured and deposited sediment was determined using the equation Mass (kg) = volume (m$^3$) x porosity (P) x sediment density. In all cases, the sediment density was assumed to be 2600 kg m$^{-3}$, the density of many aluminosilicate rocks and coastal sediments [34,35]. The calculation was applied to two different porosity scenarios. In the "standard case" the porosity of the scoured and eroded sediment was assumed to be 0.4, a commonly used porosity for coastal sediments. In the "conservative case," it was hypothesized the scoured sediment was more tightly packed (lower porosity, P = 0.37) than

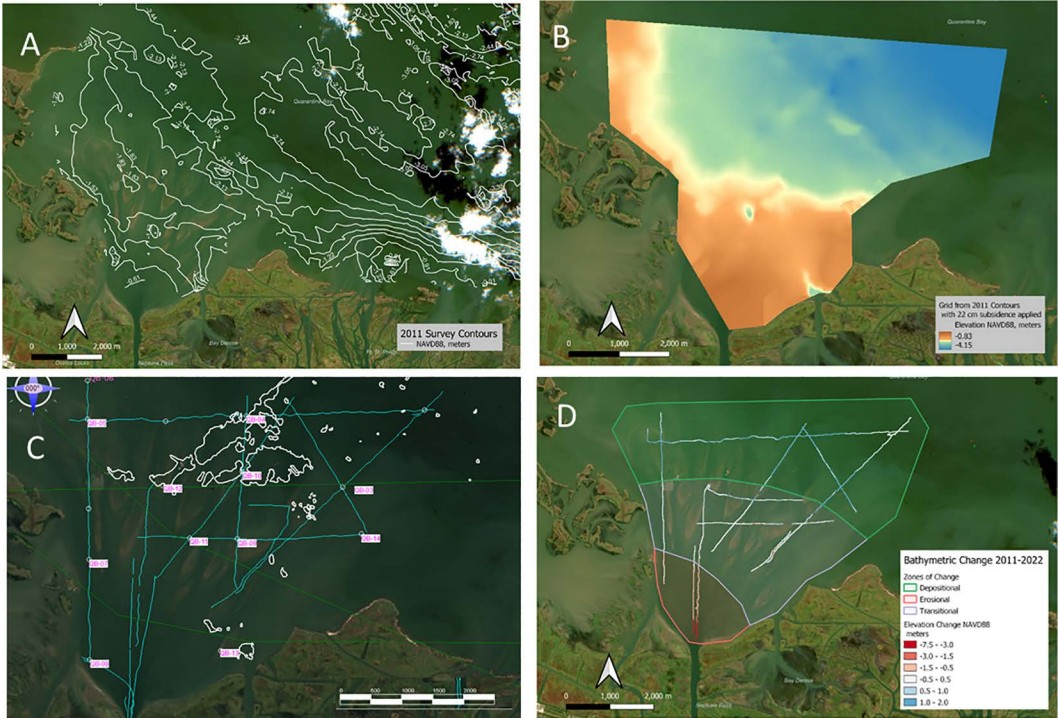

**Fig 10. Information used for the development of the sediment budget.** A and B. The 2011 survey of Quarantine Bay, expressed as contours (A) and as a color-coded grid (B). C. The core locations with the oyster reef highlighted. D. The difference map between the 2011 and 2022 surveys with the erosional, transitional, and depositional zones used in the sediment budget highlighted. Background Image Source: Copernicus Sentinel-2 Data Service. Image Date; October 24, 2022.

typical, whereas the recently deposited sediment was less tightly packed (higher porosity, P = 0.45). These values come from a survey of the literature on geotechnical properties of sediments in deltas and northern Gulf of Mexico sediments. [34–37]. This assumption would provide a conservative view on the amount of sediment that could be deposited. Using both cases allows one to understand the range of scenarios in which sediment was deposited.

## 2.9 Permitting approval

Permits were required for collection of the Vibracores in Quarantine Bay. We obtained a permit from the US Army Corps of Engineers via Louisiana's Coastal Protection Authority. That permit, P20230036, was approved on February 22, 2023. Permits were not required for the sonars or aerial drone surveys, as they were non-invasive and did not require the collection of physical materials. The aerial drone surveys were conducted by a certified drone pilot and followed Federal Aviation Authority (FAA) procedures.

## 2.10 Ethics statement

Since the project did not involve the use of human subjects the use of an Institutional Review Board (IRD) was waived. Since the project did not involve the use of vertebrate or cephalopod animals, the use of an Institutional Animal Care and Use Committee (IACUC) or equivalent ethics committee was waved.

## 3 Results

### 3.1 Discharge surveys

Table 1 and Fig 11 present the discharge in the Mississippi River and Neptune Pass during surveys conducted on May 24, 2022 and February 16, 2024. During the first survey, the discharge at Neptune Pass was 3,360 $m^3s^{-1}$, the discharge in the Mississippi River at Belle Chasse was 22,080 $m^3s^{-1}$, and the discharge in the Mississippi River above Neptune Pass was 20,290 $m^3s^{-1}$. (For reference, flood protection measures limit the flow of the lower Mississippi to a maximum of 35,400 $m^3 s^{-1}$.) The flow in the Mississippi River downstream of Neptune Pass/ Fort St Philip (i.e., the "Olga Reach") was 12,320 $m^3s^{-1}$. By difference, this suggests the amount of flow heading eastward, through Neptune Pass, Fort St Philip and other nearby crevasses was between 6,700 $m^3s^{-1}$ and 7,970 $m^3s^{-1}$. Overall, the discharge in Neptune Pass accounted for 15% to 17% of the flow in the Mississippi River, while the total eastward flow amounted to 30% to 36% of the Mississippi River. The survey conducted on February 16, 2024 indicated a discharge of 3,205 $m^3s^{-1}$ at Neptune Pass, 22,055 $m^3 s^{-1}$ in the Mississippi River at Belle Chasse, and 14,142 $m^3s^{-1}$ in the Mississippi River below the Fort St Philip Crevasse complex. In early 2024, Neptune Pass's discharge was about 15.4% of the Mississippi River at Belle Chasse which

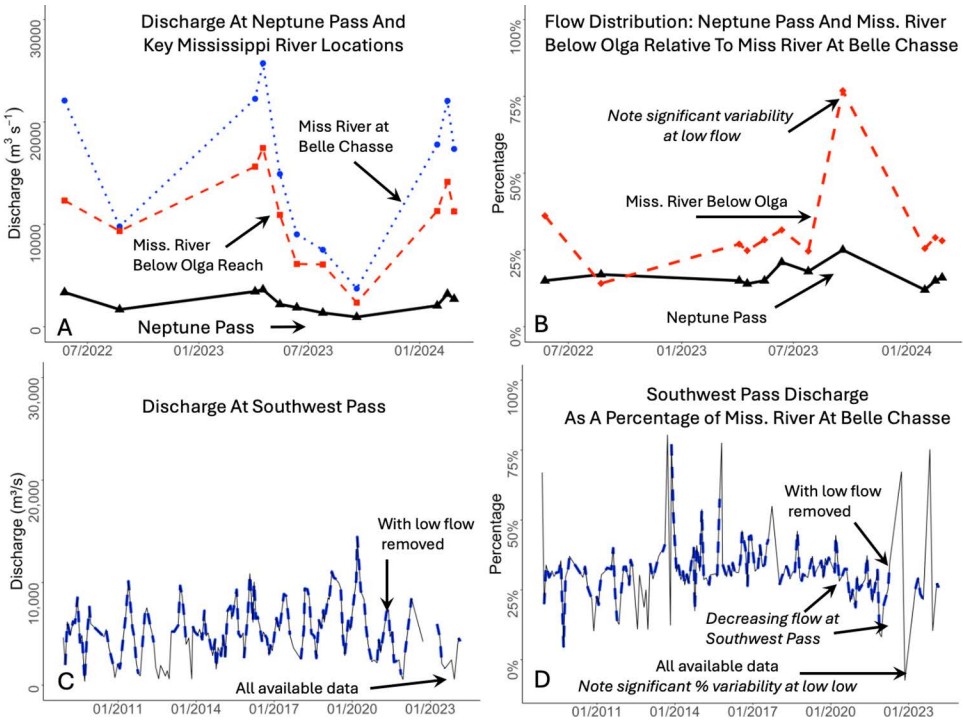

**Fig 11. Discharge in the lower Mississippi River, Neptune Pass, and nearby systems.** A. Discharge in the Mississippi River at Belle Chasse, in Neptune Pass, and in the Mississippi River at the Olga upstream and downstream locations. B. Percent of water from the Mississippi River discharged through Neptune Pass, and through the Mississippi River downstream of Neptune Pass, Fort St. Philip and the nearby crevasse, i.e., Olga Downstream. C. Discharge through Southwest Pass (the main shipping channel), with all available data, and with low flow removed. D. Percent of the Mississippi River discharged through Southwest Pass, with all available data, and with low flow removed. Note that the magnitude and percent of water flowing through Southwest Pass, the river's final distributary and main shipping channel, begins to decline in 2020, roughly contemporaneous with the expansion of Neptune Pass. To account for the impacts of extreme droughts in summer 2022 and 2023, the data is also presented with the low flow data removed. Data sources include US Geological Survey, US Army Corps of Engineers, and article authors.

revealed the water loss between Belle Chasse and Neptune Pass was about 29% of the Mississippi River at Belle Chasse.

This project also compiled data collected by the US Army Corps of Engineers on discharge at Neptune Pass and other distributaries of the lower Mississippi River, with surveys occurring on a quasi-monthly basis between August 23, 2022 and February 27, 2024 (Fig 11; S1 Supplemental Material). According to this dataset, the discharge at Neptune Pass reached a low of 950 $m^3s^{-1}$ on September 19, 2023, which occurred during a period of prolonged low flow in the Mississippi River. The highest measured flow at Neptune Pass was 3,630 $m^3s^{-1}$ on April 17, 2023 when the river was at a medium high flow of 25,726 $m^3 s^{-1}$. In this dataset, Neptune Pass accounted for between 14.5% and 25.4% of the discharge of the Mississippi River at Belle Chasse. (NB: There were times when the percentage of water discharged through Neptune Pass was greater than 25%. However, there were periods of low flow when small magnitude fluctuations in discharge had a large impact on the percentage of water discharged. These periods, like those in late summer 2023 are likely not indicative of typical discharges in the system.) The percentages of water flowing through Neptune Pass relative to Belle Chasse during the two February 2024 surveys were slightly less than the percentage during the May 24, 2022 surveys. This suggests efforts to control the flow at Neptune Pass have been successful, at least for the near term. This research cannot eliminate the possibility for the flow to change in the future, which is explored further in the discussion section.

To understand the potential for Neptune Pass to impact flow in other parts of the Mississippi River, this study developed a time series comparing the discharge at Southwest Pass (the river's main shipping channel and largest distributary) to the discharge at Belle Chasse for the period 2008–2024. These data are presented for all days with available data, with low flow values removed, where low flow is operationally defined as < 7,110 $m^3s^{-1}$, which is approximately the flow required for a salt wedge to develop in the river (Galler and Allison 2008). These data indicate the discharge at Southwest Pass was relatively stable at about 30–35% of the flow of Belle Chasse between 2008 and 2020. Beginning in 2020, the percentage of water discharged through Southwest Pass started to decline to about 18–30% of the Belle Chasse discharge. While there is some variability in this time series, and the droughts of 2022 and 2023 impacted parts of this period, the overall pattern is relatively persistent and roughly corresponds in time with the development of Neptune Pass.

## 3.2 Bathymetry

The multi-beam bathymetric survey (Fig 11) from 2022 shows evidence of significant enlargement of the Neptune Pass channel. One noteworthy feature is the presence of a thin ridge connecting the Mississippi River to Neptune Pass. This ridge has two major depths: it is about 25 m deep on the downstream side and about 15 m on the upstream side. Other noteworthy features include the presence of a large hole, about 30 m deep and 200 m wide directly downstream of the entrance to Neptune Pass, steep channel walls, and a secondary deep area (~25 m) in the center of the Neptune Pass channel. All these features are indicative of an active erosional setting, and thus broadly consistent with the rapid erosion observed from satellite/Google Earth images.

Fig 6 shows the single-beam bathymetry of Quarantine Bay overlaid on a Copernicus Sentinel-2 image of the region. The image shows a deep area near where Neptune Pass debouches into Quarantine Bay, a fan shaped delta comprised of alternating high and low grounds, about 2–5 km from Neptune Pass's mouth. This network of mouth bars and channels have lengths on the scale of $10^2$ m, widths on the scale of $10^0$ to $10^1$ m, and elevations on the order of 1–2 m. These islands and shallow water environments eventually grade into deeper water -- particularly towards the northeast in this dataset which likely denotes the edge

of the Quarantine Bay Delta deposit. Overall, the tops of the mouth bars are about -0.1 to -1.0 m NAVD 88, with negative values indicating area below a geological surface that approximates sea level which roughly corresponds to the lower intertidal/high subtidal zone.

### 3.3  CHIRP sub-bottom profiles

Fig 5 illustrates how CHIRP sub-bottom data can image below the seafloor and provide insight to the geometry and thickness of the sedimentary deposit. The figure indicates the presence of numerous mouth bars that are about 40–80 m wide and 1–2 m thick; a scale found to be consistent with the true-color satellite images, single-beam bathymetry, and SDB (Figs 4, 6, and 10). The CHIRP data also help reveal patterns of scour and deposition, as determined by the apparent removal of stratigraphic layers (for erosional surfaces), and the presence of bars and overlapping sequences (for depositional surfaces). The patterns observed were used to determine bay-wide zones of erosion, deposition, and transition in the sediment budget.

### 3.4  Aerial drone imagery

The aerial drone LiDAR imagery is presented in two ways: true color images that incorporate a digital elevation model (DEM+RGB), and reflectivity images which are color-coded by the intensity of the LiDAR signal return (Fig 7). The former is useful for understanding interactions between patterns of land growth and elevation change; parameters that influence the stability of wetlands and their capacity to respond to sea level rise [38]. The latter provides information that helps elucidate the relationship between wetland morphology and vegetation which is also a critical parameter governing wetland development [9,20,25]. In both datasets the horizontal resolution of the images is <1m; in the case of the DEM, the elevation control is < 10 cm. The reflectivity data is presented on an arbitrary scale from 0 to 1, with 0 being complete signal absorption and 1 being complete signal reflection.

The data collected at Bay Denesse and the DU Terraces provide clear examples of developing land (Fig 7). The Bay Denesse digital elevation model with red, green, and blue colors (DEM+RGB) image shows two deltas. One delta is roughly in the center of the bay and has a distinct fan-shaped morphology; another lies to its north and is more oblong-shaped with some fan-shaped offshoots visible. In the central delta, the DEM reveals differences in elevation; channel edges are generally high ground, and splay interiors are generally low ground. Additionally, the reflectivity data show that the high grounds tend to have greater reflectivity values, likely indicating a relatively dry plant community which yields stronger signal returns. Patterns are generally similar in the northern delta, but not as distinctive given the oblong shape of the delta.

In the region around the DU Terraces, a different pattern appears. An elongated and growing sand bar is present bounded by the Ostrica Channel flow path of the left (western) side and Neptune Pass on the right (eastern) side. The DEM+RGB and reflectivity data indicate the edges of the bar-system are relatively high ground. These data, when coupled with Google Earth and Copernicus Sentinel-2 data, indicate the area interior to this developing bar has been vegetating for a period of about 3 years. Additionally, the DEM+RGB data also reveals wildlife (e.g., alligators and wading birds) in high enough resolution to identify individual organisms, which raises the prospect for future research to utilize this drone technology to survey wildlife in deltas. Overall, these data indicate how the emerging technologies of high-resolution LIDAR DEM+RGB and reflectivity can be used to provide fine-scale resolution of deltaic ecogeomorphology.

## 3.5 Satellite derived bathymetry

Fig 8 shows the regression used to develop the satellite-derived bathymetry (SDB). The relationship between water depth and pixel intensity in the red band is best described by a quasi-asymptotic relationship, in which shallower water is correlated with increasing pixel intensity. This follows from the assumption that in shallow water, light from the sand bars is better reflected to space, relative to deep water, where light is less well reflected to space. The regression's limits were defined by the depth the vessel could navigate to conduct a survey on the shallow side, and the area of the main delta deposit on the deep side. Very deep water, like that near Neptune Pass's mouth in Quarantine Bay, has been removed from this regression as light reflection/depth dynamics are likely to be governed by qualitatively different factors.

Fig 12 presents the SDB for the Quarantine Bay domain. It shows a broad field of mouth bars and shallow deposits interspersed with a network of channels. The inset of the image delineates the depth along a marked pathway. Overall, the image displays the range of dimensions found in the mouth bars: hundreds to thousands of meters long, tens to hundreds of meters wide, and 1–2 meters in elevation. The findings are consistent with the in-situ sonar and CHIRP measurements, as well as the progression of satellite images (Figs 4 and 6).

## 3.6 Sediment cores

A total of 12 sediment cores were collected from Quarantine Bay. The cores were collected across the bay, in a variety of water depths, and with a range of surficial sediments (Fig 9, Table 2). The elevation of the top of the sediment cores ranges from +19 to -146 cm NAVD88. The cores span from 151 cm to 405 cm in length. Compaction in the cores ranges from -0.1% to 5.7%, with the negative value indicating expansion -- which was only found in 1 core. Overall, these core extractions are relatively typical of Vibracoring efforts in the Mississippi River Delta.

Sediments in the cores are mostly siliciclastic sediments which consist of a combination of both fine (i.e., silt and clay-sized) and coarse (i.e., sand-sized) sediment (Fig 13). These

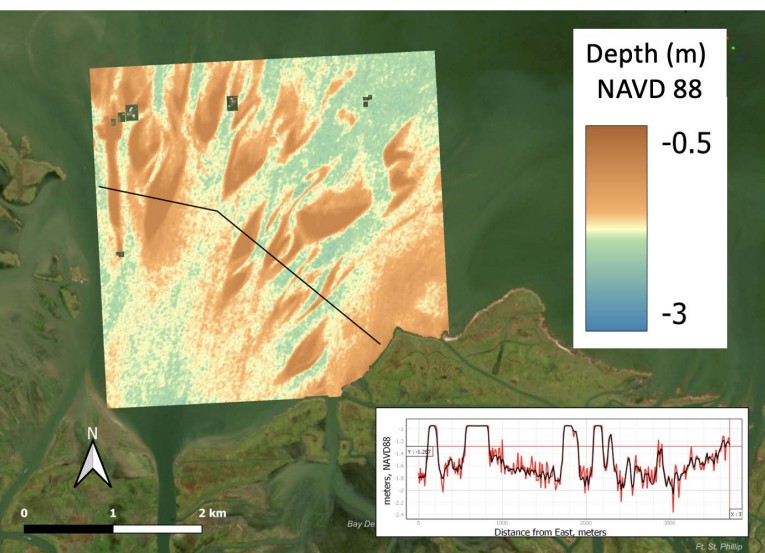

**Fig 12. Satellite-derived bathymetry of Quarantine Bay and its delta.** The linear bathymetric profile in the lower right-hand corner corresponds to the line that traverses across the delta in the main image. Background Image Source; Copernicus Sentinel-2 Data Service. Image source October 24, 2022.

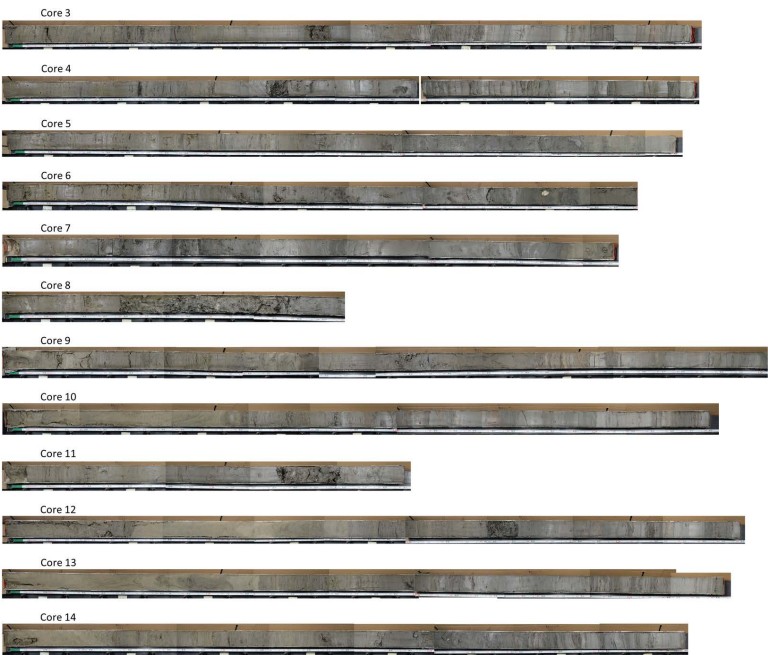

**Fig 13. Photographs of the sediment cores collected in this study.**

sediments are typically found in interbedded material, with bedding existing on scales of $10^1$ cm. Such arrangements are common in accreting deltaic deposits in the Mississippi River Delta.

Several features of the cores sampled indicate recent sediment deposition. For example, cores 9, 11, 14, and 10 have oxidized (yellow and brownish) sands near the surface. The presence of oxidized sediments suggests recent deposition, as sediments in the Mississippi River Delta's organic rich and often low-oxygen environments tend to yield reduced (black and gray) sediments. Also of particular interest is the presence of multiple, overlapping oyster shells in cores 11 (at 135–160 cm) and 9 (at 220 cm). These cores likely reflect the oyster reef present in the 2011 survey conducted by LDWF, and subsequent burial of these reefs by Neptune Pass derived sediments (Fig 10).

### 3.7 Sediment budget calculations

Table 3 presents the sediment budget which compares the amount of material scoured from the Neptune Pass channel to the amount of material deposited in the Quarantine Bay Delta. This budget indicates that a total of 6.1 x $10^6$ m³ of sediment was scoured from Neptune Pass, while a total of 1.1 x $10^7$ m³ of sediment was deposited in Quarantine Bay. Overall, this indicates a net deposition of 4.8 x $10^6$ m³ in Quarantine Bay. Converting these volumes to mass, and assuming no difference in the porosity between the scoured and deposited sediments ("standard case", section 3.6) indicates a net deposition of 7.5 x $10^9$ kg. Converting these volumes to masses, and assuming lower than usual scoured porosities and higher than usual deposited porosities ("conservative case", section 3.6) indicates a net deposition of 5.6 x $10^9$ kg deposition. Using differing values for the porosity of sediments in Neptune Pass and Quarantine Bay provides a means of bracketing the range of scenarios for deposition and erosion. Overall, results indicates that under the standard case, the deposit in Quarantine Bay is between 56% and 79% greater than would be expected from scour in Neptune Pass alone,

**Table 3. Sediment budget estimates for Neptune Pass and Quarantine Bay.**

**Estimate of Excavation in Neptune Pass**

| Date | Mean Channel Area | Channel Length | Channel Volume |
|---|---|---|---|
| | (m²) | (m) | (m³) |
| 2016 ADCP | 3.17 x 10² | 2.63 x 10³ | 8.34 x 10⁵ |
| 2016 above Transducer | – | – | 2.56 x 10³ |
| 2022 Multibeam | – | – | 6.95 x 10⁶ |
| | | **Net Erosion** | **-6.12 x 10⁶** |

**Estimate of Deposition in Quarantine Bay**

| Zone | Mean Seafloor Change | Area | Net Volume Change |
|---|---|---|---|
| | (m) | (m²) | (m³) |
| Erosional | -0.81 | 2.9 x 10⁶ | -2.3 x 10⁶ |
| Transitional | -0.12 | 9.8 x 10⁶ | -1.2 x 10⁶ |
| Depositional | 1 | 1.4 x 10⁷ | 14 x 10⁶ |
| **Total** | | 2.7 x 10⁷ | **11 x 10⁶** |

**Estimate of Net Deposition in Quarantine Bay**

| Location | Volume | Density | Typical Case Porosity | Typical Case Mass | Conservative Case Porosity | Conservative Case Mass |
|---|---|---|---|---|---|---|
| | (m³) | kg/m³ | % | kg | % | kg |
| Channel Scour | -6.1 x 10⁶ | 2600 | 0.4 | -9.5 x 10⁹ | 0.37 | -1.0x 10¹⁰ |
| Bay Deposition | 1.1 x 10⁷ | 2600 | 0.4 | 1.7 x 10¹⁰ | 0.45 | 1.6 x 10¹⁰ |
| **Net Deposition** | **4.8 x 10⁶** | | | **7.5 x 10⁹** | | **5.6 x 10⁹** |

Sediment budget for Neptune Pass and Quarantine Bay Delta.

with the lower value corresponding to the conservative case and the higher value corresponding to the standard case. (NB: These value round to 60% to 80% if fewer significant figures are used.)

## 4 Discussion

### 4.1 Neptune Pass discharge in perspective

The magnitude of water discharged through Neptune Pass is significant in a regional context, and potentially a global context as well. During medium and high flow Neptune Pass's discharge is larger than the average discharge of the 10th largest river in the United States, the Missouri River (Q ~2,400 m³ s⁻¹). Neptune Pass' discharge is similar to the average flow of the Rhine River (Q ~2,900 m³ s⁻¹), and thus comparable to discharge of the ~ 100th largest river on Earth. Regionally, Neptune Pass is the largest new offshoot of the Mississippi River to develop in nearly a century. Its size is analogous to the Wax Lake Outlet, an offshoot of the Atchafalaya River (a distributary of the Mississippi River) that was created via dredging in 1941 [18,19], and which is often considered one of the best modern systems to study river delta development [19,20,39]. Neptune Pass's discharge is also roughly equivalent to the channels that feed the several ~ 200 km² deltas in the Birdsfoot region of the Mississippi River Delta, such as Cubit's Gap (~3,000 m³ s⁻¹). However, Neptune Pass is smaller than the anthropogenic crevasse created in the Mississippi River levee near Caernarvon, LA (9,250 m³ s⁻¹) during the 1927 flood [40]. That crevasse, which was closed after the flood, is among the largest of crevasses to occur in recorded history. Overall, these comparisons indicate that Neptune Pass is a large, and in the modern era, unusual event.

The discharge of Neptune Pass at high flow is greater than the maximum planned discharge of the Mid-Barataria Sediment Diversion, which aims to shunt up to 2,000 $m^3$ $s^{-1}$ of water and its associated sediment load into central Barataria Bay. That project's goal is to rebuild and sustain existing wetlands by mimicking the natural processes that originally built the Mississippi River Delta in an embayment that lost about 30% of its land area over the past century [16,17]. The Mid-Barataria Sediment Diversion will cost about $3 billion and will create about 50–70 $km^2$ of land over the next 50 years, making it one of the largest and most expensive environmental projects in US history. Since Neptune Pass carries more water than the Mid-Barataria Sediment Diversion and developed without any direct intervention, it is potentially a large and low-cost coastal restoration option. However, any land-building benefits should be viewed in the context of the safe management of the Mississippi River. Changes to river currents and the development of shoals and sandbars downstream of Neptune Pass could create hazards to navigation in one the country's largest commercial shipping corridors [21,22]. Given the range of possible impacts of Neptune Pass, decisions about its future should be based on the best available science and analysis, and this article is among the first significant efforts to provide such information.

Neptune Pass is part of a broader region where about 30–39% of the Mississippi River flows eastward (Fig 11). Indeed, the eastward flow between Ostrica Locks and Olga Downstream (i.e., below Fort St Philip) can approach the flow through Southwest Pass -- which has long been the largest distributary of the Mississippi River. For example, on January 30, 2024 the eastward flow in this reach was 4,537 $m^3$ $s^{-1}$, while the discharge of Southwest Pass on January 31, 2024 was 4,593 $m^3$ $s^{-1}$. The values were similarly close on April 4 and 5, 2023 when the eastward flow was 5,978 $m^3$ $s^{-1}$, and the discharge at Southwest Pass was 5969 $m^3$ $s^{-1}$. Furthermore, the amount of water lost from the river between Belle Chasse and Olga downstream, i.e., eastward flux through Neptune Pass, Fort St Philip, and other nearby channels, is significant relative to the amount of water flowing in the main channel downstream of the Olga Reach. During non-drought periods, the eastward flow was 36–57% as large as the downstream flow (roughly 1/3–1/2 as large). Taken holistically, these data suggest the recent development of Neptune Pass along with the longer-term evolution of Fort St. Philip that began expanding after a large flood in 1973, may constitute a partial avulsion of the Mississippi River. In other words, this is one of the largest changes to the largest river in the United States in nearly a century.

## 4.2  Causes of Neptune Pass development

Few instances of large-scale crevasse splays development have been studied during the modern era of observational geosciences -- in which tools like hydroacoustics and remote-sensing allowed for detailed and precise measurements of coastal morphodynamics. This era, roughly the 20th and 21st centuries, also corresponds to a period in which large coastal engineering projects limited the development of avulsions, bifurcations, crevasse splays, and related features. For instance, the most cited example of a new delta in the Mississippi River Delta is the Wax Lake Delta [18,19,28,39]. However, the channel of the Wax Lake Outlet was dredged in 1941 and the delta was not observed until years following the large flood in 1973, meaning that over 30 years passed between channel formation and delta development. Furthermore, a dearth of regular satellite images in the 1970s and 1980s meant the early-stage development of the Wax Lake Delta was understudied. There are a few examples of new deltas forming in other environments. Satellite images have shown deltas emerging in Greenland and other areas with retreating ice sheets, while analyses of historical maps revealed the development of new deltas near the mouth of the Yellow and Mississippi Rivers over the past several centuries [2,12,41,42]. Filling this research gap is important. The processes that drive crevasse splay

initiation and delta growth is a critical need for the coastal research community, given that many deltas are prone to submerge this century due to accelerating sea level rise, increased subsidence, wetland loss and other human impacts to coastal shorelines [2,5,10,43].

Remotely sensed images indicate Neptune Pass rapidly expanded after early 2019, with the initial phase of expansion complete by 2021. Figs 2–4 indicate the channel expanded by about an order of magnitude in one or two years, making it the most rapid channel expansion to occur since the crevasses of the 1927 flood [40]. The multibeam survey [Fig 3] indicates evolution likely continued until 2022, when rocks were placed near the mouth of Neptune Pass. The erosional ridge at the entrance to Neptune Pass, the large scour hole, and steep channel walls all indicate an active erosional environment (Fig 3 and 4).

While a comprehensive examination of the causes of Neptune Pass' development would require additional research and analysis, data in this study in combination with other publicly available data, point to five non-mutually exclusive hypotheses for the system's rapid expansion.

**Extended periods of high river water.** The period from 2008 to 2020 had multiple extended high-water events, including 7 openings of the Bonnet Carre Spillway, a measure enacted when the lower Mississippi River reaches its maximum allowable flow (rivergauges.com). The region also experienced at least 3 seasons when the Mississippi River's flow approached, but did not reach, the threshold for a Bonnet Carre Spillway opening (rivergauges.com). Multiple years of high flow would have put pressure on the Proto-Neptune Pass system, creating erosive forces that could have led to its rapid expansion.

**Degrading infrastructure.** The levee systems along the east bank of the Mississippi River south of Bohemia, Louisiana are not maintained. While some channel stability measures are present, such as rock walls and revetments, this region is prone to crevassing. Earlier crevasses near Neptune Pass include the roughly 12 crevasses near Fort St. Philip that developed between about 1973 and 2000, Mardi Gras Pass which opened in 2011 or 2012, and the crevasse by Ostrica Locks that expanded between 1998 and 2010 [44,45]. In these cases, pressure from the Mississippi River has overtaken geotechnically weak areas, resulting in crevasses. Without artificial levees along the Mississippi River, crevassing would be common, and regularly occurred in the years before human-made levees were in place [1,46].

**Hurricane impacts.** Hurricanes can cause storm surges that travel up the Mississippi River. The increased flow can put pressure on channel walls, which can contribute to crevassing. During Hurricane Barry (July 12–13, 2019), a small (~ 0.3 m) storm surge up the Mississippi

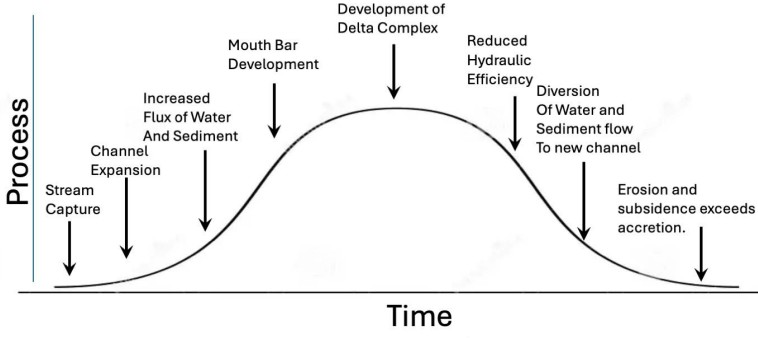

**Fig 14. The temporal progression of the delta cycle, modified from Figure 2 in [18] - Roberts (1997).**

River (rivergauges.com) occurred when the river was at high flow (~ 28,500 m$^3$ s$^{-1}$). This was within 1 year of the rapid expansion of Neptune Pass, and the temporal coincidence is intriguing. However, Barry's surge was substantially smaller than surges during Hurricanes Katrina (~5m; 2005) and Isaac (~2.5 m; 2012), which did not appreciably impact other crevasses near Neptune Pass.

**Backstepping of the Mississippi River Delta.** Some scholars suggest the mouth of the Mississippi River is moving inland, a result of the combined impacts of rising global sea levels and subsiding lands [47]. This hypothesis leads to the prediction that crevasses would develop upstream of the river's current mouth as the delta "backsteps" and retrogrades landward. The development of Neptune Pass and other nearby crevasses, such as the Fort St. Philip Crevasse Complex and Mardi Gras Pass are consistent with this theory of deltaic backstepping.

**Faulting.** Geological faults are present below the ground in Louisiana. One study from 2014 shows faults near Neptune Pass [48]. It is possible these faults created a weakness or low elevation point in the river's levee that became vulnerable to erosive forces of the Mississippi River.

While the aforementioned factors are not mutually exclusive and could have acted synergistically to increase the size of Neptune Pass, article authors hold that the combination of years of high flow and degrading infrastructure are the most likely causes of the expansion. High river flows would have placed intense forces on the Proto-Neptune Pass, while infrastructure deterioration likely created structural weaknesses. It is intriguing that Hurricane Barry's storm surge in the Mississippi River occurred slightly before Neptune Pass's expansion. However, Barry's surge was relatively modest and short-lived, particularly when compared to the duration of river floods. The backstepping hypothesis, which theorizes the Mississippi River Delta is moving landward as relative sea level rises, has been discussed in Louisiana's coastal community for several years [47]. While results presented herein are consistent with retrograding, further study is needed to definitively link Neptune Pass's expansion to the backstepping hypothesis. Finally, insufficient evidence exists in the published literature to thoroughly evaluate the faulting hypothesis -- though the available evidence is not inconsistent with it.

## 4.3  Development of Neptune Pass in the context of deltaic development

**4.3.1  The delta cycle.**  The development of Neptune Pass and the Quarantine Bay Delta can be understood in the context of the geological literature on deltaic development (Fig 14). One key element is the "delta cycle" which envisions deltaic development as a cyclical process of channel and splay growth and decay [18]. The cycle starts with a crevasse in a river channel that rapidly expands. This expansion is governed by a positive feedback loop in which rapidly flowing water erodes the channel, drawing more water. This increases the erosive power of the channel, which causes further crevasse expansion and increases water and sediment discharge into an open bay. As more water flows into an open bay, less water flows through the old channel.

Over time, sediment input to the open bay results in landform development, typically forming a complex that includes mouth bars, natural levees, mudflats, and marshes [18]. In later stages of the delta cycle the flux of water and sediment becomes restricted as landforms develop to their maximum size, resulting in less sediment input and reduced marsh accretion [12,18,38]. During these late phases the channel becomes less hydrodynamically efficient, resulting in more water flow in the original channel, creating pressure that eventually leads to the formation of a new crevasse. This restarts the delta cycle elsewhere [18].

The overall spatial patterns associated with the delta cycle are relatively consistent across the Mississippi River Delta, though the temporal and spatial scale of these systems vary considerably. Small channels tend to have a discharge of about 50–500 m s$^{-1}$ and create splays

with areas of about $10^{-1}$ to $10^{1}$ km$^2$ that last decades [12,24]. Medium-sized channels have discharges of about 500–5,000 m$^3$ s$^{-1}$ and create deltas with areas of $10^1$ to $10^2$ km$^2$ that last decades to centuries [12]. The largest channels have discharges 5,000–50,000 m$^3$ s$^{-1}$ and create delta lobes consisting of multiple smaller deltas with areas of $10^3$ to $10^4$ km$^2$ that last centuries to millennia [12,24]. Given its discharge, Neptune Pass is the size of a system that would have -- under pre-industrial conditions -- built hundreds km$^2$ of land and lasted for centuries.

The development of Neptune Pass is comparable to parts of the delta cycle, including:

**Rapid channel expansion.** Delta cycle theory predicts new crevasses should rapidly expand, a result of a positive feedback loop between channel discharge and channel scour [18]. The rapid expansion of Neptune Pass in 2019 and 2020 follows theoretical predictions.

**Mouth bar development.** The development of mouth bars in Quarantine Bay is consistent with the land-building phase of delta development [18,30,49,50]. The islands observed in Quarantine Bay are morphologically similar to mouth bars in other well-known developing deltas, such as the Wax Lake Delta today, and early 20th century maps of the Cubit's Gap Delta [39,51].

**Reduced hydrodynamic efficiency in the main channel** Shoaling in the Mississippi River observed downstream of Neptune Pass [21,22] is consistent with delta cycle theory noting a reduction in hydrodynamic efficiency in the main channel as flow is diverted into a newly developing crevasse [18].

**4.3.2 Sediment dynamics at river mouth.** The development of landforms associated with Neptune Pass are also consistent with geomorphological theory developed for river mouths [30,49,51]. The geomorphology of sediment-rich rivers that discharge into shallow basins is governed, in large part, by friction. (NB: While the Mississippi River carries less sediment than it did 100 years ago, it still carries in > 100 x 10$^6$ metric tons of sediment per year, and is thus, relative to many other rivers, sediment rich [34].) Closest to the river mouth where velocities are high, crevasses have a zone with no deposition or erosion (Fig 15 modified after [49]). Further from the river mouth, the flow spreads out and velocities decrease, inducing sedimentation. The heaviest, coarse-grain material settles out first, creating a bar that is often tear-drop shaped (Fig 15 modified after [49]). The interiors of these bars tend to receive less sediment, and the sediment they do receive tends to be relatively fine-grained. The result is an area of poorly consolidated mud and open water in the zone between the natural levees [20,30, 49].

This pattern of river mouth development envisioned by Wright [49] is apparent in the Neptune Pass data. Most notably, the Copernicus Sentinel-2 imagery of Quarantine Bay, the satellite-derived bathymetry of Quarantine Bay, and the aerial drone LiDAR imagery of Bay Denesse, all show tear-dropped shaped islands -- as predicted by geomorphological theory (Figs 4, 6–7, and 12). The three images with explicit elevation data (aerial drone and SDB- Figs 7, 12, and 15) indicate that mouth bar edges are higher than the interiors. Morphologically, this pattern in elevation looks similar to other developing deltas like the Wax Lake Delta, the Cubit's Gap Delta, and the delta in the Davis Pond Freshwater Diversion [18,20,52,53]. The use of satellite-derived bathymetry in this study, which we believe is the first case for deltas, provides a new way by which deltaic distributary deposits and channels can be investigated. This can potentially lead to further high-resolution studies in some of the most dynamic coastal environments.

The return flow of sediments, after their discharge in a bay, is another potentially import-ant geomorphic pathway for sediment accumulation. This pathway was best described for the Wax Lake Delta in southwest Louisiana [54]. The authors of [54] hypothesized that some sediments carried by the main conveyance channel (i.e., the Wax Lake Outlet) initially dis-charged beyond the delta, can be transported to posterior regions by strong onshore winds. This contributes to sediment accumulation in inshore marshes.

A. Flow Regime: Friction Dominated River Mouth

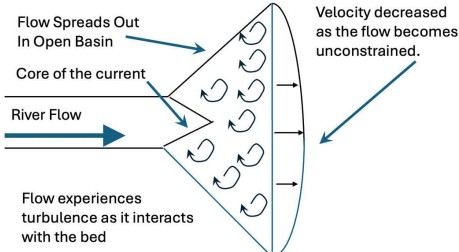

B. Geomorphology Friction Dominated River Mouth

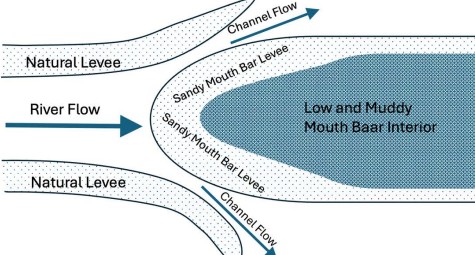

C. Bay Denesse: LIDAR DEM+RGB

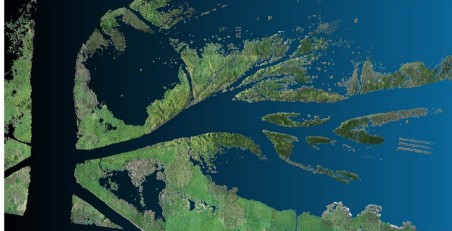

D. Bay Denesse: LIDAR Reflectivity

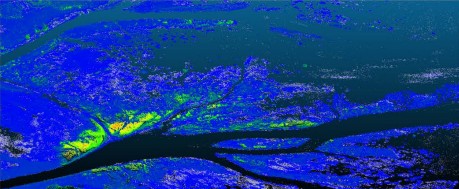

**Fig 15. Development of a mouth bar.** A. Hydrodynamic flow patterns and relative velocities at a friction-dominated river mouth Modified after Figure 2 in [49]. B. Patterns of sediment accumulation in a sediment-rich, friction-dominated river mouth. Modified Figure 5 in [49]. C. LiDAR true color and DEM imagery of the Bay Denesse Delta. D. Reflectivity of the Bay Denesse Delta. Note how the theoretical patterns of mouth bar development in B are similar to the actual shapes that develop in the Bay Denesse Delta.

The impacts of return flow are apparent in the Ducks Unlimited terraces and the accreting sandbar northwest of the mouth of Neptune Pass. A combination of pre-2020 Google Earth imagery, the aerial drone imagery, and Sentinel-2 images reveals a zone of accretion (Figs 3, 4, 7, 12, and 16). Areas of open water have become vegetated, and the sandbar is elongating at a rate of >100 m yr$^{-1}$. Article authors propose that sediments are ejected from Neptune Pass into the northeastward direction, and then returned eastward via winds and waves, where they accumulate in marshes.

**4.3.3 The role of vegetation.** Vegetation can also play an important role in deltaic development because stems and shoots can slow water flow which traps sediments,

while roots and rhizomes can promote accretion [9,20,55]. As such, the development of vegetated areas is not simply a mark of stable land, but also a key factor in its development.

Vegetation appears to be taking root in the Bay Denesse Delta and the Ducks Unlimited terraces. As indicated by both aerial drone data and on the ground observation, the distribution of vegetation in Bay Denesse matches patterns from ecogeomorphological theory (Fig 16). For example, the reflectivity image of the Bay Denesse Delta shows greater reflectivity on delta edges relative to the delta interior, and field observations (A. Kolker pers observation) indicate that these high grounds include plants like Roseaucane (*Phragmites australis)* and Black Willow (*Salix nigra*). Such plants are commonly found in developing deltas [20,56]. Reflectivity values are lower in the bar interior, and field observations indicate these areas are dominated by plants like arrowhead *(Sagittaria latifolia)*. Additionally, reflectivity data points to submerged aquatic vegetation, most likely widgeon grass (*Ruppia maritima*), in some areas on anterior and lateral edges of the Bay Denesse Delta. This pattern of elevation and vegetation is broadly consistent with geomorphic theory of higher ground at the edges of mouth bars and lower ground in the middle. It is also consistent patterns observed in other developing deltas in Louisiana like the Wax Lake Delta, the Cubit's Gap Delta, and the West Bay Mississippi River Diversion Delta [20,50,57,58]. As such, the deltaic features being built by Neptune Pass appears to follow ecogeormorphic patterns similar to those in historic and natural deltaic wetlands in coastal Louisiana.

## 4.4  Sediment budget interpretation and landscape evolution

The results from the sediment budget help to address one important question about the Neptune Pass/Quarantine Bay system: is the system building land? More specifically, are the emerging lands in Quarantine Bay "new land," material derived from the Mississippi River. Alternatively, are these lands composed of simply "redistributed sediment," material scoured from the erosion of Neptune Pass's main channel?

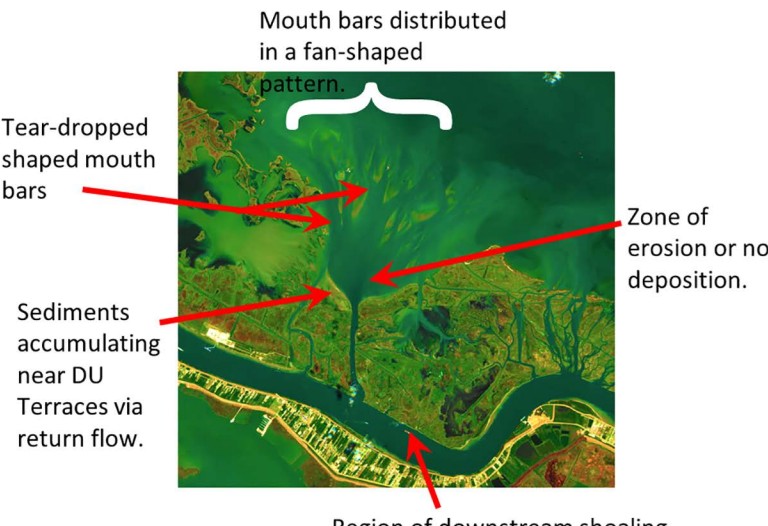

**Fig 16.  The Quarantine Bay Delta, annotated with major deltaic features noted.** These features follow classical geomorphological models include the dynamics of river mouths [49] and the delta cycle [18]. Background image: Copernicus Sentinel-2 Data Service. Image date: October 24, 2024.

This question can be addressed through a sediment volume and mass balance. In this approach, the volume (or mass) of sediment removed from Neptune Pass can be compared to the volume (or mass) of sediment that has been deposited in Quarantine Bay. If the quantity of newly-deposited sediments in Quarantine Bay exceeds the volume of material extracted, this would support the "new land" hypothesis because it implies a large input of sediment from somewhere else, and the Mississippi River is the only significant source. If the volume deposited is less that the volume scoured, it would support the "redistributed" hypothesis because it suggests sediment input from the Mississippi River has little net significant impact on land development -- at least at the spatial and temporal scales measured here.

Results from the sediment budget indicate a net deposition of $4.8 \times 10^6$ m$^3$ of sediment in Quarantine Bay (Table 3). Using standard estimates of sediment porosity and volume, this indicates $7.5 \times 10^9$ kg of sediment was deposited in Quarantine Bay. Under the more conservative scenario (i.e., assuming that eroded material was less dense than usual and deposited material was more dense than usual), an estimated $5.6 \times 10^9$ kg of sediment was deposited in Quarantine Bay. Overall, these results indicate the Neptune Pass/Quarantine Bay system is net depositional. Given the geography of the region, this positive balance of sediment is almost certainly coming from the Mississippi River.

These findings have implications for coastal management. They suggest that the Neptune Pass/Quarantine Bay system functions similarly to the sediment diversions that are part of Louisiana's coastal restoration strategies [17], which aim to mimic the fluvial-deltaic processes that created the Mississippi River Delta and its smaller subdeltas [18,24,46]. As described above, the Quarantine Bay Delta, the Bay Denesse Delta, and parts of the Ducks Unlimited terraces appear to be following well-documented pathways of deltaic sediment transport, deposition, and land growth [18,30,34,39,52,54]. Most importantly, Neptune Pass is leading to the development of new land in Louisiana, and very likely at a lower cost than the Mid-Barataria and Mid-Breton Sediment Diversions. However, article authors caution that the environmental benefits of building land should, from a management perspective, be judged in light of the potential for changing currents and conditions in the Mississippi River to affect safe navigation.

There are a few caveats to these findings. First, the "new land" vs "redistributed land" hypotheses are not entirely mutually exclusive. It is theoretically possible, and even likely, that sediments deposited in Quarantine Bay could be derived from both channel scour and the Mississippi River. Indeed, the sediment budget provides a net-accounting of sediment rather than detailed analysis of every transport pathway. More detailed field studies and physics-based numerical models may well provide additional useful information on the sediment transport pathways of the Neptune Pass/Quarantine Bay system.

## 4.5  The future of Neptune Pass

The future status of Neptune Pass is unknown and depends on a range of factors including its behavior under current day conditions that are mostly unregulated, and future conditions that may include additional control structures to reduce discharge. The US Army Corps of Engineers has proposed plans to reduce the flow of Neptune Pass with a combination of a rock structure at the confluence of the river and the pass, and by adding sediment retention devices in Quarantine Bay [22]. In theirs view, such actions are necessary to maintain safe navigation in the Mississippi River - a major commercial pathway in the United States. Here we use geomorphic theory to examine the future of the Neptune Pass/Quarantine Bay system, with and without the US Army Corps of Engineers's plans.

Overall, the future of the Quarantine Bay Delta is likely to be governed by a balance between multiple factors that include: sediment input, the efficiency of sediment trapping, patterns of vegetation growth, wind-driven waves, and relative sea level rise [18]. Factors that could promote delta growth include high sediment input and high rates of sediment trapping --

which can be augmented by vegetation. High sediment input would result from large inflows from the Mississippi River, while strong sediment trapping would result from robust growth of the plant community. Factors that could slow or reverse the rate of delta growth include a wave climate that results in erosion, reduced sediment input, low rates of sediment trapping, slow vegetation growth, and high rates of relative sea level rise. The erosional wave climate could result from strong storms including cold fronts (which occur ~20 times per year between October and May), and hurricanes (which are stronger but occur less frequently). The reduced rate of sediment trapping would result from a slow growing plant community, or ineffective sediment retention devices, and the high rates of relative sea level rise would result from accelerating climate change and/or high rates of subsidence. which are common but poorly quantified in this section of the Mississippi River.

One parameter affecting the influence of these factors is the delta's position in the tidal frame if/when flow control measures are implemented. A relatively high position in the tidal frame could promote vegetation growth which could lead to a positive feedback loop of land growth [20,59]. Under this scenario, a high elevation could promote vegetation growth, which would enhance sediment trapping, leading to wetland accretion that create sediment trapping landforms [9,20,53]. On the other hand, a relatively low position in the tidal frame could lead to a positive feedback loop that hinders land building. Under this scenario, a low elevation could promote the formation of wind-driven waves which would erode landforms, reduce vegetation growth and sediment trapping, allowing subsidence and sea level rise to continue to lower the system's place in the tidal frame [38].

In general, reducing the input of water and sediment input -- as proposed by the US Army Corps of Engineers- would hinder wetland development in the Quarantine Bay Delta by decreasing wetland accretion, while allowing waves and relative sea-level rise to become proportionately more important. However, sediment trapping structures -- like those proposed by the US Army Corps of Engineers [22] could increase the potential for wetland accretion and land development. The full balance between these contrasting actions deserves further research that includes both enhanced observations and numerical modelling. The full impacts of any plan should be viewed in the context the sustainable management of the region including coastal restoration, flood protection, and navigation in the Mississippi River.

Perhaps the most significant broader questions regarding the future of Neptune Pass is whether similar events could occur elsewhere. Indeed, there have been other crevasses in the region in recent decades, including Mardi Gras Pass (2011; Q ~300–600 $m^3 s^{-1}$), Ostrica (~2000; Q ~300–600 $m^3 s^{-1}$) and the Fort St Philip Crevasse Complex (1972~1980; Q ~2,000 ~4,000 $m^3 s^{-1}$ [45], Fig 1). The factors initiating the initiation of these crevasses likely are the same ones that lead to the formation of Neptune Pass: years of high river flow in a region of aging infrastructure. Satellite images and field observations suggest that the there are other weak spots in the lowermost Mississippi River that could be prone to crevassing/expansion. Reducing the flow of water through Neptune Pass could, at least in theory, increase the pressure in the river, creating other crevasses elsewhere in the system. Given the importance of the Mississippi River as an economic driver for the U.S., supporting communities and industries up and down the river, this incident at Neptune Pass highlights the need for a more comprehensive view of the lowermost Mississippi River and the network of engineered and self-organizing (i.e., natural) systems that influence water systems.

## Supporting information

**S1 Supplemental Material. Discharge Data in The Mississippi River.** Data from 5/24/2022 and 2/16/2022 are from these authors while the other data are from the US Army Corps of Engineers routine surveys and made available by distribution from Eden Krolopp and

colleagues. In some cases, surveys were conducted over 2-day period, and here the first day of that survey is listed). The units for discharges are $m^3 s^{-1}$.
(DOCX)

## Acknowledgements

We thank Richie Blink from Delta Discovery Tours, and Mike Brown from University of New Orleans for vessel support. We thank Anne Patton from With Bells On LLC for help preparing this manuscript, and we thank Wyatt Oconnell for help with data management.

## Author contributions

**Conceptualization:** Alexander S Kolker, H. Dallon Weathers, Christy Swann, Alisha A. Renfro.

**Data curation:** Alexander S Kolker, H. Dallon Weathers, Christy Swann.

**Formal analysis:** Alexander S Kolker, H. Dallon Weathers, Christy Swann.

**Funding acquisition:** Alexander S Kolker, Christy Swann, Alisha A. Renfro.

**Investigation:** Alexander S Kolker, H. Dallon Weathers, Christy Swann, Alisha A. Renfro.

**Methodology:** Alexander S Kolker, Christy Swann.

**Project administration:** Alexander S Kolker, Alisha A. Renfro.

**Resources:** Alexander S Kolker, Christy Swann, Alisha A. Renfro.

**Software:** Christy Swann.

**Supervision:** Alexander S Kolker.

**Validation:** Alexander S Kolker, H. Dallon Weathers.

**Visualization:** Alexander S Kolker, H. Dallon Weathers, Christy Swann.

**Writing – original draft:** Alexander S Kolker, H. Dallon Weathers, Christy Swann.

**Writing – review & editing:** Alexander S Kolker, Alisha A. Renfro.

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
