## [Decision Letter · Decision Letter 0]

26 Aug 2024

PONE-D-24-23965Distributary development in a 21st century river: The evolution of Neptune Pass and its delta, the largest new offshoot of the Mississippi RiverPLOS ONE

Dear Dr. Kolker,

Thank you for submitting your manuscript to PLOS ONE. After careful consideration, we feel that it has merit but does not fully meet PLOS ONE’s publication criteria as it currently stands. Therefore, we invite you to submit a revised version of the manuscript that addresses the points raised during the review process.

We look forward to receiving your revised manuscript.

Kind regards,

Venkatramanan S, Ph.D.

Academic Editor

PLOS ONE

Journal Requirements:

   "This project was funded, in part, through a sub-contract with the Louisiana Coastal Protection and Restoration Authority, who was funded under Award No. GNTCP18LA0035 from the Gulf Coast Ecosystem Restoration Council (RESTORE Council), and through multiple contracts with the National Wildlife Federation. This work represents contract #20220831Task Order No.4 from the Louisiana Coastal Protection And Restoration Authority to the Louisiana Universities Marine Consortium. The data, statements, findings, conclusions, and recommendations are those of the authors and do not necessarily reflect any determinations, views, or policies of the RESTORE Council. We acknowledge that Dr. Alisha Renfro from the National Wildlife Federation contributed to this manuscript in her capacity as a scientist and scholar. Links to project sponsors can be found here: https://coastal.la.gov;
https://www.restorethegulf.gov;
https://www.nwf.org"

5. In the online submission form, you indicated that "All data are available from the Louisiana Coastal Protection And Restoration Authority, by making a data request here:https://cims.coastal.louisiana.gov/DataRequest.aspx"

8. We note that Figures 1-4, 6-10, 12,15 and 16 in your submission contain map/satellite images which may be copyrighted. All PLOS content is published under the Creative Commons Attribution License (CC BY 4.0), which means that the manuscript, images, and Supporting Information files will be freely available online, and any third party is permitted to access, download, copy, distribute, and use these materials in any way, even commercially, with proper attribution. For these reasons, we cannot publish previously copyrighted maps or satellite images created using proprietary data, such as Google software (Google Maps, Street View, and Earth). For more information, see our copyright guidelines: http://journals.plos.org/plosone/s/licenses-and-copyright.

a. You may seek permission from the original copyright holder of Figures 1-4, 6-10, 12,15 and 16 to publish the content specifically under the CC BY 4.0 license.  

Additional Editor Comments:

The study primarily relies on conventional methods for analyzing sediment transport and land formation, such as sediment core collections and remote sensing, which may not capture more complex aspects of distributary development and delta dynamics. Additionally, while it investigates sediment dynamics in Neptune Pass, it lacks a deeper exploration of the broader environmental or hydrological impacts, missing a nuanced understanding of how these processes affect local and regional ecosystems. The results are presented with varying levels of detail, which complicates full interpretation, as sediment volumes are reported with different levels of confidence and assumptions, potentially affecting the reliability of the conclusions. The discussion on potential management strategies is somewhat generic, lacking specific recommendations or actionable insights that could directly inform policy or management practices. Moreover, the paper does not adequately compare its findings with other studies or historical data, which would help contextualize the significance of Neptune Pass’s development within a broader framework of delta evolution and river management. The study reports a net deposition of sediment in Quarantine Bay but does not fully account for possible variations in sediment density or other factors influencing the sediment budget, which could strengthen the findings. The discussion of the "new land" versus "redistributed sediment" hypotheses may oversimplify the sediment dynamics, as both processes might contribute to land development, warranting a more detailed examination of sediment sources and pathways. Future predictions and management interventions are discussed without specific, data-driven scenarios; incorporating more detailed modeling and field studies could better predict the impacts of reduced flow or sediment retention measures on Neptune Pass. The generalizations about coastal management implications lack site-specific factors or challenges, and more precise recommendations tailored to the local context would be beneficial. Lastly, while the study highlights Neptune Pass as a net land-building system, it does not thoroughly explore the ecological consequences, such as impacts on local flora and fauna or changes in habitat conditions, which would provide a more comprehensive understanding of the delta's evolution.

Reviewers' comments:

Reviewer's Responses to Questions

**Comments to the Author**

1. Is the manuscript technically sound, and do the data support the conclusions?

Reviewer #1: Yes

2. Has the statistical analysis been performed appropriately and rigorously? 

Reviewer #1: N/A

3. Have the authors made all data underlying the findings in their manuscript fully available?

Reviewer #1: Yes

4. Is the manuscript presented in an intelligible fashion and written in standard English?

Reviewer #1: Yes

5. Review Comments to the Author

Reviewer #1: Excellent paper that has left no ground unturned. Will be useful to both delta geoscientists and delta managers. Well done!

Minor points to rectify:

Line 118: potentially leading to ...

Lines 122ff and 126ff: Where do these hypotheses come from? If you are presenting them here for the first time then reword this as the presentation looks awkward and gives the impression that that these hypotheses are already extant.

Line 145: "to generate (rather than provide in order to avoid using "provide" twice in the same sentence).

Line 146ff: This information should also be useful to delta geoscientists, so although this is mentioned above, it can be reiterated following the management bit for delta geoscience management elsewhere.

Line 809: Fig. 14. Need to specify that this delta cycle refers to the Mississippi and reference Roberts as reference 18.

Line 584: "also consistent with ..."

Figures: Figure 1 is a bit disappointing. For an international readership, please show a regional USA map with the Mississippi delta in inset and add geographical coordinates to the figure.

6. PLOS authors have the option to publish the peer review history of their article (what does this mean? ). If published, this will include your full peer review and any attached files.

**Do you want your identity to be public for this peer review?** For information about this choice, including consent withdrawal, please see our Privacy Policy .

Reviewer #1: **Yes: ** Edward Anthony

---

## [Author Response · Author response to Decision Letter 1]

5 Dec 2024

Dear PLOS One:

Please see this file that contains a response to the comments provided by the reviewer, editor, and journal. We believe the comments these individuals provided greatly improved the quality of the manuscript, and we hope that it now meets PLOS One's standards for publication.

Should you have further questions, please do not hesitate to ask me.

My co-authors and I look forward to working with PLOS One.

Sincerely,

-Alexander S. Kolker

Response To Reviewer:

Reviewer #1: Excellent paper that has left no ground unturned. Will be useful to both delta geoscientists and delta managers. Well done!

> Thank you!

Minor points to rectify:

Line 118: potentially leading to ...

Done. Please see line 119 in the tracked changes version.

Lines 122ff and 126ff: Where do these hypotheses come from? If you are presenting them here for the first time then reword this as the presentation looks awkward and gives the impression that that these hypotheses are already extant.

This has been addressed. Please see lines 124- 125 in the tracked changes version.

Line 145: "to generate (rather than provide in order to avoid using "provide" twice in the same sentence).

This has been addressed. Please see line 151 in the tracked changes version.

Line 146ff: This information should also be useful to delta geoscientists, so although this is mentioned above, it can be reiterated following the management bit for delta geoscience management elsewhere.

Yes. We agree that this information should be useful to a broad range of geoscientists. We now note this in lines 152-153 of the tracked changes manuscript.

Line 809: Fig. 14. Need to specify that this delta cycle refers to the Mississippi and reference Roberts as reference 18.

This has been addressed. Please see line 873 of the tracked changes version.

Line 584: "also consistent with ..."

This has been addressed (We believe the reviewer was actually referring to line 846)

Please see line 911-912 of the tracked changes version.

Figures: Figure 1 is a bit disappointing. For an international readership, please show a regional USA map with the Mississippi delta in inset and add geographical coordinates to the figure.

Thank you for this comment. We have fixed Figure 1 following the suggestions of this reviewer. We hope it now satisfies this reviewer.

Response to Editor's Comments:

Editor Comment 1. The study primarily relies on conventional methods for analyzing sediment transport and land formation, such as sediment core collections and remote sensing, which may not capture more complex aspects of distributary development and delta dynamics.

> We thank the editor for thoughtful comment, and we appreciate the editor's desire to understand the complex aspects of distributary development. However, we used the methods we used because they work. We think scientific methods should be judged based on whether they are appropriate to address specific questions at hand, rather than whether they are novel.

We appreciate the editor's interested in seeing new methods. We turn their attention to satellite-derived bathymetry, and the use high-resolution LIDAR (ie. Sections 3.6 and 3.6 and Figs 7, 8 and 12. We now highlight how this study pioneers the use satellite-derived bathymetry in a river delta. See lines 937-940 of the tracked changes version.

Our study also advances the technology of LIDAR DEM+RBG and reflectivity in river deltas Section 3.5 and Fig 7). We now highlight how this technology can be used in river deltas, where it's use has previously been limited. Please see lines 650-652 of the tracked changes version.

Editor Comment 2.

Additionally, while it investigates sediment dynamics in Neptune Pass, it lacks a deeper exploration of the broader environmental or hydrological impacts, missing a nuanced understanding of how these processes affect local and regional ecosystems.

> While we very much appreciate the editor's desires to see a discussion of the broad hydrological impacts of Neptune Pass, we believe the comment that our study lacks this material is incorrect. We dedicate a significant amount of text to describing Neptune Pass's broader environmental impacts. Key examples of places where describe the environmental impacts of the system include:

Section 5.1: Presents a detailed discussion of the local, regional and global implications of Neptune Pass. (Lines 729 - 777 of the tracked changes manuscript)

Section 5.3.1, and 5.3.2 Presents a description of the development of Neptune Pass and the Quarantine Bay Delta in the context of geomorphic theory. (Lines 864 - 959 of the tracked changes manuscript)

Section 5.3.3. Presents a discussion of development of deltas in the context of local vegetation. (Lines 965-986 of the tracked changes manuscript)

Section 5.4. and 5.5. Presents a detailed discussion of the development of Neptune Pass and the Quarantine Bay Delta in the context of regional environmental planning. (Lines 1014-132, and 1034-1109 of the tracked changes manuscript)

Editor Comment 3

The results are presented with varying levels of detail, which complicates full interpretation, as sediment volumes are reported with different levels of confidence and assumptions, potentially affecting the reliability of the conclusions.

> We appreciate the reviewer's desire to see the results with a more specific level of detail. We clarify here that both the scour estimates and deposition estimates are guided by marine sonars, and thus have a strong degree of comparability. To further clarify how the sediment volumes were calculated, we added an additional sentence that reads, "Using differing values for the porosity of sediments in Neptune Pass and Quarantine Bay provides a means of bracketing the range of scenarios for deposition and erosion." See lines 712-714 of the tracked changes version.

Editor Comment 4

The discussion on potential management strategies is somewhat generic, lacking specific recommendations or actionable insights that could directly inform policy or management practices.

> We thank the editor for asking to see additional material on management. To address this comment, we clarified and expanded Section 5.5, which examines the management implications of this system, and pathways for future management related research. Please see the entirety of Section 5.5, lines 1034-1109 of the tracked changes version.

Moreover, the paper does not adequately compare its findings with other studies or historical data, which would help contextualize the significance of Neptune Pass’s development within a broader framework of delta evolution and river management.

> While we very much appreciate the editor's desire to see historical data and comparisons to other systems, this material is already in the paper. Please see:

Section 5. 1 Neptune Pass Discharge in Perspective:

*We compare the size of Neptune Pass to other crevasses/river locally, nationally and globally. Please see lines 729-744 of the tracked changes version.

* We compare the size of Neptune Pass to one of the largest environmental restoration projects in the United States, the Mid Barataria Sediment Diversion. Please see lines 746-760 of the tracked changes version.

Section 5.3 Development of Neptune Pass in the context of deltaic development.

Section 5.3.1. The delta cycle.

*We compare Neptune Pass to well established theory on delta development.

Please see lines 863-908 of the tracked changes version.

* We compare Neptune Pass to well established theory on the sediment dynamics at river mouths. Please see lines 911-969 of the tracked changes version.

5.3.3 The role of vegetation.

* We compare Neptune Pass to well established ecogeomorphologic theory. Please see lines 965-986 of the tracked changes version.

The study reports a net deposition of sediment in Quarantine Bay but does not fully account for possible variations in sediment density or other factors influencing the sediment budget, which could strengthen the findings.

> The density of the sediments is not likely to vary significantly. Many siliciclastic sediments in the Mississippi River Delta, such as quartz (i.e. sand), feldspar, and clay minerals have a density that is close to 2600 kg/m3. such, many authors use this value in their calculations- see for example Blum and Roberts (2012)- citation 34. As such, we do not believe that any additional calculation is needed, or even appropriate.

However, there could be differences in the amount of pore space in the sediments. As such, we used two porosity scenarios in our sediment budget. The first calculation used a sediment porosity of 0.4- the value most commonly used in the literature [34]. In the second calculation we assumed the porosity of original sediments was less that typical (ie more tightly compacted that usual), and that the porosity of the deposited sediments was greater than typical (i.e. less tightly compact than usual) - using literature values. This provided a reasonable estimate of the range of porosities that could be exist in this system. We thus believe our methods specifically address this reviewer's concerns. This information is reflected in lines 501-513 of the tracked changes version.

The discussion of the "new land" versus "redistributed sediment" hypotheses may oversimplify the sediment dynamics, as both processes might contribute to land development, warranting a more detailed examination of sediment sources and pathways.

>We agree with the reviewer that both processes might contribute to land development, which is why we have specifically addressed this issue in the text. We refer the editor to lines 134 to 135 in the tracked changes version which reads, "(It should be noted that these hypotheses are not mutually exclusive and represent end-members on a spectrum of sediment input possibilities.". Please see lines 134-135 of the tracked changes version.

We also refer the editor to lines 1026-1032 which reads, "There are a few caveats to the aforementioned findings. First, the "new land" vs "redistributed land” hypotheses are not entirely mutually exclusive. It is theoretically possible, and even likely, that sediments deposited in Quarantine Bay could be derived from both channel scour and the Mississippi River......." Please see line 1026-1032 of the tracked changes version.

Future predictions and management interventions are discussed without specific, data-driven scenarios; incorporating more detailed modeling and field studies could better predict the impacts of reduced flow or sediment retention measures on Neptune Pass.

> We agree with the reviewer that more modelling studies are important. However, we believe that those are best left for future studies that build on this work. Following one of the editor's earlier remarks, we do now conclude the paper will a call for further research, and we hope that this call helps address the editor's concerns. Please see lines 1085-1093 of the tracked changes version.

The generalizations about coastal management implications lack site-specific factors or challenges, and more precise recommendations tailored to the local context would be beneficial.

>We have expanded the section on coastal management to include more site-specific material. We refer the editor to lines 1034-1108.

Lastly, while the study highlights Neptune Pass as a net land-building system, it does not thoroughly explore the ecological consequences, such as impacts on local flora and fauna or changes in habitat conditions, which would provide a more comprehensive understanding of the delta's evolution.

We thank the reviewer for their interest in the ecology of the region. We point out that In section 5.3.3, "The role of vegetation", our study describe the role of vegetation in Neptune Pass and its associated deltas. Our paper also refers the reader to the classical papers on ecogeomorphological theory for developing deltas in coastal Louisiana. While we would love to see more papers on the ecology of this system, those are best done by other manuscripts that could be submitted to this journal at another point in time. Please see lines 965-986 of the tracked changes version.

Journal Comments

> This has been addressed. To the best of our knowledge, this manuscript meets the standards of PLOS One.

> We have added Section 3.9 Permitting Approval to the manuscript. This section describes the permits needed and provides details on the permits that were obtained.

> All funding related information has been removed from the manuscript. Of particular relevance- the material that was in section 6.0 Acknowledgements has been deleted so as to conform with journal policies.

"This project was funded, in part, through a sub-contract with the Louisiana Coastal Protection and Restoration Authority, who was funded under Award No. GNTCP18LA0035 from the Gulf Coast Ecosystem Restoration Council (RESTORE Council), and through multiple contracts with the National Wildlife Federation. This work represents contract #20220831Task Order No.4 from the Louisiana Coastal Protection And Restoration Authority to the Louisiana Universities Marine Consortium. The data, statements, findings, conclusions, and recommendations are those of the authors and do not necessarily reflect any determinations, views, or policies of the RESTORE Council. We acknowledge that Dr. Alisha Renfro from the National Wildlife Federation contributed to this manuscript in her capacity as a scientist and scholar. Links to project sponsors can be found here: https://coastal.la.gov;
https://www.restorethegulf.gov;
https://www.nwf.org"

If this statement is not correct you must amend it as needed. Please include this amended Role of Funder statement in your cover letter; we will change the online submission form on your behalf.

> Thank you for asking for this clarification, our previous statement has been clarified. Our cover letter now includes the following text " We acknowledge that Dr. Alisha Renfro from the National Wildlife Federation contributed to this manuscript in her capacity as a scientist and scholar. In particular, some of her insights on the geology of the Mississippi River Delta were part of the discussion section. Louisiana's Coastal Protection And Restoration Authority, the project's other funder, had no role in study design, data collection and analysis, decision to publish, or preparation of the manuscript "

5. In the online submission form, you indicated that "All data are available from the Louisiana Coastal Protection And Restoration Authority, by making a data request here:https://cims.coastal.louisiana.gov/DataRequest.aspx"

This policy applies to all data except where public deposition would breach compliance with the protocol approved by your research ethics board. If your data cannot be made publicly available f

---

## [Decision Letter · Decision Letter 1]

19 Jan 2025

PONE-D-24-23965R1Distributary development in a 21st century river: The evolution of Neptune Pass and its delta, the largest new offshoot of the Mississippi RiverPLOS ONE

Dear Dr. Kolker,

Thank you for submitting your manuscript to PLOS ONE. After careful consideration, we feel that it has merit but does not fully meet PLOS ONE’s publication criteria as it currently stands. Therefore, we invite you to submit a revised version of the manuscript that addresses the points raised during the review process.

We look forward to receiving your revised manuscript.

Kind regards,

Venkatramanan S, Ph.D.

Academic Editor

PLOS ONE

Journal Requirements:

Reviewers' comments:

Reviewer's Responses to Questions

**Comments to the Author**

1. If the authors have adequately addressed your comments raised in a previous round of review and you feel that this manuscript is now acceptable for publication, you may indicate that here to bypass the “Comments to the Author” section, enter your conflict of interest statement in the “Confidential to Editor” section, and submit your "Accept" recommendation.

Reviewer #1: (No Response)

Reviewer #2: All comments have been addressed

2. Is the manuscript technically sound, and do the data support the conclusions?

Reviewer #1: Yes

Reviewer #2: Yes

3. Has the statistical analysis been performed appropriately and rigorously? 

Reviewer #1: N/A

Reviewer #2: N/A

4. Have the authors made all data underlying the findings in their manuscript fully available?

Reviewer #1: Yes

Reviewer #2: Yes

5. Is the manuscript presented in an intelligible fashion and written in standard English?

Reviewer #1: Yes

Reviewer #2: Yes

6. Review Comments to the Author

Reviewer #1: Review of PLOS ONE

“Distributary development in a 21st century river: The evolution of Neptune Pass and its delta, the largest new offshoot of the Mississippi River” by A. Kolker et al.

I reiterate my satisfaction with this manuscript which has monitored the potential inception factors, growth, and dynamics of a new delta pass with significantly rapid lobe sedimentation. I am sure it will be an impactful contribution to a better understanding of the Mississippi River delta and to delta science in general.

I SUGGEST MINOR EDITS THAT ARE LISTED BELOW:

Delta and delta lobe: The authors systematically refer to “delta” development with regards to the sediment accumulation associated with Neptune Pass. This is somewhat unfortunate as the background context is the Mississippi river and its DELTA. It may be less confusing to refer to delta LOBES (which all these other components of the Mississippi DELTA are), rather than individual DELTAS, e.g., lines 96-98, 121, 425-6. Sub-delta (line 699) is another new addition that does not clarify the picture.

• Lines 41-42: “observation, coincides with an era when rivers have been controlled by large engineering projects”. Delete comma after observation and be less sweeping regarding river modification: “many rivers have been controlled ...”.

• Lines 48-49: “it is building is comprised .. Awkward statement.

• Line 66: “..and often serve as a locus for human activities”. Use a more recent and more pertinent reference by some of the same authors:

- Anthony, E., Syvitski, J., Zăinescu, F., Nicholls, R.J., Cohen, K.M., Marriner, N., Saito, Y., Day, J., Minderhoud, P.S.J., Amorosi, A., Chen, Z., Morhange, C., Tamura, T., Vespremeanu-Stroe, A., Besset, M., Sabatier, F., Kaniewski, D., Maselli, V., 2024. Delta sustainability from the Holocene to the Anthropocene and envisioning the future. Nature Sustainability, 7,1235-1246. https://doi.org/10.1038/s41893-024-01426-3

• Line 69: and alter the biogeochemistry and ecology of coastal waters [7–9]. Consider also the recent paper on tipping points in delta channels:

- van de Vijsel, R.C., Scheffer, M., Hoitink, A.J.F., 2024. Tipping points in river deltas. Nature Reviews Earth & Environment, 5, 843–858. https://doi.org/10.1038/s43017-024-00610-5

• Line 131-2: “It should be noted that these hypotheses are not mutually exclusive and represent end-members on a spectrum of sediment input possibilities”. Remove parentheses as this is an important point.

• Line 173: “It is across the river ..”. Quaint or colloquial expression. Please rectify.

• Line 223: “by placing ‘a’ rock wall ..”

• Line 225: “to inform comments on such this proposal …” Rectify.

• Line 243: “and its potential impacts ‘on’… (on, in lieu of to).

• Lines 251-2: “data on discharge in the Mississippi River using data from multiple outside sources”. Awkward. Reword as “data on discharge in the Mississippi River and from multiple outside sources”

• Line 255: “data was compiled”. Use rather “data were compiled”. Check manuscript and homogenize, as, sometimes, “data” is/are referred to in the singular (in abstract for instance : “there is relatively little data …) or plural form (line 149: “data that provide..”; line 290: “survey data were integrated ..”) form.

• Line 1069: Syvitski J, Anthony E, F. SYZ,. Check author name SYZ….

Reviewer #2: I would like to commend the authors for their efforts in conducting this research to understand the role of new distributary systems in larger rivers in developing landforms through sedimentation processes. The methodology adopted for this study is clear and can be applied to other parts of the world where new distributary systems are being planned. This research also enhances our understanding of erosional and accretional processes along coastlines influenced by new river distributary systems. Additionally, the authors have significantly improved the article based on the constructive comments and suggestions provided by reviewers and editors during the previous review process. I recommend that this article be accepted for publication.

7. PLOS authors have the option to publish the peer review history of their article (what does this mean? ). If published, this will include your full peer review and any attached files.

**Do you want your identity to be public for this peer review?** For information about this choice, including consent withdrawal, please see our Privacy Policy .

Reviewer #1: **Yes: ** Edward Anthony

Reviewer #2: No

---

## [Author Response · Author response to Decision Letter 2]

3 Feb 2025

We thank the reviewer for these comments. They were helpful, and we appreciate the reviewer's comments. Please see our detailed responses below.

•Delta and delta lobe: The authors systematically refer to “delta” development with regards to the sediment accumulation associated with Neptune Pass. This is somewhat unfortunate as the background context is the Mississippi river and its DELTA. It may be less confusing to refer to delta LOBES (which all these other components of the Mississippi DELTA are), rather than individual DELTAS, e.g., lines 96-98, 121, 425-6. Sub-delta (line 699) is another new addition that does not clarify the picture.

The reviewers' point is well taken. In this version, we use the term, "delta," when we speak of the generic geological feature. We use a, "D," when we refer to a specific named feature, such as the Mississippi River Delta." We have removed the term subdelta because we recognize that it could be confusing in some situations. See for example lines 96, 101, 769, 954.

• Lines 41-42: “observation, coincides with an era when rivers have been controlled by large engineering projects”. Delete comma after observation and be less sweeping regarding river modification: “many rivers have been controlled ...”

This has been completed. Please see line 41 of the track changes version.

• Lines 48-49: “it is building is comprised .. Awkward statement.

This statement has been revised. Please see line 48 of the tracked changes version.

• Line 66: “..and often serve as a locus for human activities”. Use a more recent and more pertinent reference by some of the same authors:

- Anthony, E., Syvitski, J., Zăinescu, F., Nicholls, R.J., Cohen, K.M., Marriner, N., Saito, Y., Day, J., Minderhoud, P.S.J., Amorosi, A., Chen, Z., Morhange, C., Tamura, T., Vespremeanu-Stroe, A., Besset, M., Sabatier, F., Kaniewski, D., Maselli, V., 2024. Delta sustainability from the Holocene to the Anthropocene and envisioning the future. Nature Sustainability, 7,1235-1246. https://doi.org/10.1038/s41893-024-01426-3

This reference has been updated as per the reviewer's helpful suggestion. Please see lines 1196-1199 of the tracked changes version.

• Line 69: and alter the biogeochemistry and ecology of coastal waters [7–9]. Consider also the recent paper on tipping points in delta channels:

- van de Vijsel, R.C., Scheffer, M., Hoitink, A.J.F., 2024. Tipping points in river deltas. Nature Reviews Earth & Environment, 5, 843–858. https://doi.org/10.1038/s43017-024-00610-5

This was a helpful suggestion, we appreciate hearing about this new paper. It is indeed quite interesting. However, we believe the current list of citations is sufficient.

• Line 131-2: “It should be noted that these hypotheses are not mutually exclusive and represent end-members on a spectrum of sediment input possibilities”. Remove parentheses as this is an important point.

This has been addressed. Please see lines 140-142 of the tracked changes version.

• Line 173: “It is across the river ..”. Quaint or colloquial expression. Please rectify.

This has been addressed. Please see lines 186-187 of the tracked changes version.

• Line 223: “by placing ‘a’ rock wall ..”

This has been addressed. Please see line 239 of the tracked changes version.

• Line 225: “to inform comments on such this proposal …” Rectify.

This has been addressed. Please see lines 241-242 of the tracked changes version.

• Line 243: “and its potential impacts ‘on’… (on, in lieu of to).

This has been addressed. Please see line 267 of the tracked changes version.

• Lines 251-2: “data on discharge in the Mississippi River using data from multiple outside sources”. Awkward. Reword as “data on discharge in the Mississippi River and from multiple outside sources”

This has been addressed. Please see lines 275-276 of the tracked changes version.

• Line 255: “data was compiled”. Use rather “data were compiled”. Check manuscript and homogenize, as, sometimes, “data” is/are referred to in the singular (in abstract for instance : “there is relatively little data …) or plural form (line 149: “data that provide..”; line 290: “survey data were integrated ..”) form.

This was addressed. Please see lines 279 and 309 of the tracked changes version.

• Line 1069: Syvitski J, Anthony E, F. SYZ,. Check author name SYZ….

This was addressed, the earlier version was missing a name. Please see lines 1196-1199 of the tracked changes version.

---

## [Editor Report · Decision Letter 2]

20 Feb 2025

Distributary development in a 21st century river: The evolution of Neptune Pass and its delta, the largest new offshoot of the Mississippi River

PONE-D-24-23965R2

Dear Dr. Kolker,

We’re pleased to inform you that your manuscript has been judged scientifically suitable for publication and will be formally accepted for publication once it meets all outstanding technical requirements.

Kind regards,

Venkatramanan S, Ph.D.

Academic Editor

PLOS ONE

Additional Editor Comments (optional):

Based on the reviewer’s comments, the authors have revised the paper, and it is now ready for publication.
---

## [Editor Report · Acceptance letter]

PONE-D-24-23965R2

PLOS ONE

Dear Dr. Kolker,

I'm pleased to inform you that your manuscript has been deemed suitable for publication in PLOS ONE. Congratulations! Your manuscript is now being handed over to our production team.

Kind regards,

on behalf of

Dr. Venkatramanan S

Academic Editor

PLOS ONE